# SOFTMAX-INDUCED ILL-CONDITIONING IN TRANSFORMER MODELS

## ABSTRACT

Transformers have become ubiquitous in modern machine learning applications, yet their training remains a challenging task often requiring extensive trial and error. Unlike previous architectures, transformers possess unique attention-based components, which can complicate the training process. The standard optimization algorithm, Gradient Descent, consistently underperforms in this context, underscoring the need for a deeper understanding of these difficulties. To understand this phenomenon, we analyze a simplified yet representative softmax attention model. Our local analysis of the gradient dynamics reveals that the Jacobian of the softmax function itself acts as a preconditioner. We show that when sufficiently many attention coefficients are small across multiple training examples, the Jacobian of the softmax becomes ill-conditioned, severely degrading the local curvature of the loss, which in turn slows the convergence of Gradient Descent. Our experiments confirm these theoretical findings on the critical impact of softmax on the dynamics of Gradient Descent.

## 1 INTRODUCTION

In recent years, transformer architectures have demonstrated remarkable success in various applications from natural language processing (Devlin et al., 2019; Achiam et al., 2023) to computer vision (Dosovitskiy et al., 2020). However, despite their impressive performance, our theoretical understanding of transformer training remains limited. For example, Stochastic Gradient Descent (SGD), which has been a staple optimization algorithm for deep learning models, fails to train transformers effectively (Liu et al., 2020). This has led to the belief that there are unique elements in the transformer architecture and the associated loss landscape that introduce challenges distinct from those in other architectures like Convolutional Neural Networks (CNNs).

Recently, several papers have attempted to explain this phenomenon by comparing SGD with adaptive methods. Indeed, empirical observations indicate that while SGD outperforms adaptive methods on tasks such as ImageNet classification with CNNs (Wilson et al., 2017), the opposite is true for transformer training. Liu et al. (2020) attributed this failure of SGD to vanishing and imbalanced gradients observed in experiments. Meanwhile, Jiang et al. (2023) introduced a variant of the condition number and empirically showed that it is larger on the path taken by SGD than the one taken by Adam for standard transformer architectures. Zhang et al. (2020) hypothesized that the performance gap might be due to the heavy-tailed and non-Gaussian nature of the stochastic gradient noise. However, Kunstner et al. (2023) challenged this explanation by showing that even in the full-batch case, SGD performs poorly compared to adaptive methods. Although these works highlight interesting features of transformer training that are different for Adam compared with SGD, their theoretical explanations of the source of these differences remain very limited.

In parallel, a different line of work focused on theoretically analyzing the dynamics of GD on transformers and providing global convergence guarantees without explaining its poor performance. For instance, Tarzanagh et al. (2023b) showed that GD on a simplified attention model solves a max-margin problem and provided guarantees for global and local convergence in the binary classification setting. Meanwhile, Abbe et al. (2023) showed that the incremental learning phenomenon, whereby singular values are learned one-by-one, is present in transformers. However, these works did not provide any rates of convergence that would shed light on the slow or lack of convergence of SGD in

practice. Others provided rates of convergence but were limited to heavily overparameterized settings (Wu et al., 2023; Deora et al.) and assumed pretrained weights Liu et al. (2023).

Clearly, there exists a gap between the current theoretical analysis of GD and the empirical observations demonstrating its difficulty in optimizing transformers. This leads to the following question:

*Why does Gradient Descent perform poorly on attention models?*

### 1.1 PAPER CONTRIBUTIONS

Our work identifies a fundamental source of this difficulty: the softmax function itself. Our analysis centers on a simplified one-layer softmax attention model that highlights the key optimization challenges while remaining tractable for theoretical analysis. Our contributions are as follows:

1. We establish sufficient conditions for global linear convergence of GD in the overparameterized regime, but demonstrate that this setting cannot explain the poor performance observed in practice due to its reliance on bounded attention weights.

2. We provide a rigorous analysis of GD dynamics in the underparameterized regime around sparse attention solutions, proving that the condition number scales quadratically with the sparsity ratio and that this ill-conditioning stems from the spectral properties of the softmax Jacobian.

3. We conduct controlled experiments on synthetic data and Vision Transformers that confirm our theoretical predictions, demonstrating clear convergence degradation as attention sparsity increases, thus providing empirical support for theoretical analysis.

**Notation.** For any integer $n \geq 1$, we denote by $[n]$ the set $\{1, \ldots, n\}$. We use lower-case and upper-case bold letters to represent vectors and matrices, respectively. The $i$-th entry of a vector $\boldsymbol{x}$ is denoted as $x_i$. We denote the Euclidean norm of a vector $\boldsymbol{x}$ by $\|\boldsymbol{x}\|_2$. We use $\sigma_{min}(\mathbf{A})$ (resp. $\sigma_{max}(\mathbf{A})$) and $\lambda_{min}(\mathbf{A})$ (resp. $\lambda_{max}(\mathbf{A})$) to denote the minimum (resp. maximum) singular value and the minimum (resp. maximum) eigenvalue of a matrix $\mathbf{A}$. We use $\Delta^n$ to denote the $n$-dimensional probability simplex in $\mathbb{R}^{n+1}$ and $\mathring{\Delta}^n$ to denote its interior. $T\Delta^n$ denotes the tangent space of the n-dimensional probability simplex. $\phi(\cdot)$ denotes the softmax function. For any vector $\boldsymbol{x} \in \mathbb{R}^n$, we use $\text{diag}(\boldsymbol{x})$ to denote the diagonal $n \times n$ matrix whose diagonal entries are $x_1, \ldots, x_n$. We use $\mathbb{1}_n$ to denote the vector of ones in $\mathbb{R}^n$. $\delta_{ij}$ denotes the Kronecker delta and is equal to 1 when $i = j$ and 0 otherwise.

## 2 PRELIMINARIES

A central component of transformers is the attention mechanism (Vaswani et al., 2017). Given input matrices $\boldsymbol{X} \in \mathbb{R}^{n \times d}$ and $\boldsymbol{Z} \in \mathbb{R}^{m \times d}$, queries, keys, and values are defined as

$$\boldsymbol{Q} = \boldsymbol{Z}\boldsymbol{W}_Q, \quad \boldsymbol{K} = \boldsymbol{X}\boldsymbol{W}_K, \quad \boldsymbol{V} = \boldsymbol{X}\boldsymbol{W}_V, \tag{1}$$

with learnable weight matrices $\boldsymbol{W}_Q, \boldsymbol{W}_K, \boldsymbol{W}_V$. The output is then $\text{Attention}(\boldsymbol{Q}, \boldsymbol{K}, \boldsymbol{V}) = \phi(\boldsymbol{Q}\boldsymbol{K}^\top)\boldsymbol{V}$, where $\phi$ is the softmax.

**Tunable tokens.** In practice, some of the query tokens are sometimes also learned. For example, a [CLS] or prompt token is typically appended to the input features of a model for the purpose of classification or to adapt the model to new tasks. The latter is referred to as prompt tuning and has been introduced as a more efficient alternative to fine-tuning the transformer weights (Lester et al., 2021; Liu et al., 2023). Furthermore, Oymak et al. (2023) identified scenarios in which prompt attention, which corresponds to freezing the self-attention weights and tuning the prompt token, is more expressive than self-attention. Following (Oymak et al. (2023); Tarzanagh et al. (2023b)), we will therefore consider a simplified attention model with one tunable token $\boldsymbol{p} \in \mathbb{R}^d$ and a value vector $\boldsymbol{v} \in \mathbb{R}^d$. The key and query weights are combined in one matrix $\boldsymbol{W} = \boldsymbol{W}_Q\boldsymbol{W}_K^\top$. This model outputs a scalar which can be used for classification or regression:

$$f(\boldsymbol{X}) = \phi(\mathbf{X}\boldsymbol{W}\mathbf{p})^\top \mathbf{X}\mathbf{v} \in \mathbb{R} \tag{2}$$

**Problem Setting.** Given a training dataset $(\boldsymbol{X}_i, y_i)_{i=1}^N$ with $\boldsymbol{X}_i \in \mathbb{R}^{n \times d}$ and $y_i \in \mathbb{R}$, we consider the empirical risk minimization problem

$$\mathcal{L}(\theta) := \frac{1}{N} \sum_{i=1}^N \ell\big(f(\boldsymbol{X}_i; \theta), y_i\big), \tag{3}$$

where $\theta = (\boldsymbol{W}, \boldsymbol{p}, \boldsymbol{v})$ collects all trainable parameters, and $\ell : \mathbb{R} \times \mathbb{R} \to \mathbb{R}$ is differentiable, $m$-strongly convex and $L$-smooth in its first argument. Without loss of generality, we assume $\ell \geq 0$ and attains its global minimum at $0$. The model is trained using Gradient Descent.

We adopt the following assumptions, which define the simplified setting for our analysis:

**Assumption A.** *We work under the following assumptions throughout:*

*(A1)* **Model Simplification.** *The weight matrix $\boldsymbol{W}$ and value vector $\boldsymbol{v}$ are fixed, and only the prompt parameter $\boldsymbol{p}$ is trained. We will thus write the loss as a function of $\boldsymbol{p}$ only:*

$$\mathcal{L}(\boldsymbol{p}) = \frac{1}{N} \sum_{i=1}^N \ell(\phi(\boldsymbol{K}_i \boldsymbol{p})^\top \boldsymbol{X}_i \boldsymbol{v}, y_i), \;\; \boldsymbol{K}_i = \boldsymbol{X}_i \mathbf{W}. \tag{4}$$

*(A2)* **Realizability / Student–Teacher.** *There exists $\boldsymbol{p}^\star \in \mathbb{R}^d$ such that for every $i \in [N]$*

$$y_i = \phi(\boldsymbol{K}_i \boldsymbol{p}^\star)^\top \boldsymbol{X}_i \boldsymbol{v}, \qquad \boldsymbol{K}_i = \boldsymbol{X}_i \boldsymbol{W}.$$

*(A3)* **Full-rank Key–Query Map.** *The matrix $\boldsymbol{W}$ is full rank.*

**Discussion.** These baseline assumptions serve to isolate the role of the softmax mechanism in shaping the optimization landscape. (A1) is the most significant: freezing $\boldsymbol{v}$ is empirically motivated by the observation that value weights ($W_V$) converge much faster than query/key parameters, so the true bottleneck lies in the softmax dynamics (Li et al., 2023b). This choice also mirrors practical regimes such as prompt tuning, where value vectors are not updated. (A2) is the standard student–teacher assumption, ensuring that the training loss can be driven to zero at $\boldsymbol{p}^\star$ without approximation error. Finally, (A3) rules out degenerate key–query maps; while multi-head architectures are necessarily low-rank, in the single-head setting it is natural to assume full rank. Together, these assumptions provide a clean baseline in which ill-conditioning phenomena can be studied in isolation from other confounding effects.

**Jacobian of the Softmax Function.** For any vector $\mathbf{z}$ in $\mathbb{R}^n$, let $\boldsymbol{J}(z)$ be the $n$ by $n$ matrix given by:

$$\mathbf{J}(\mathbf{z}) = \mathrm{diag}(\mathbf{z}) - \mathbf{z}\mathbf{z}^\top = \mathrm{diag}(\mathbf{z})(\mathbf{I}_n - \mathbb{1}_n \mathbf{z}^\top), \tag{5}$$

then the Jacobian of the softmax function $\phi : \mathbb{R}^n \to \Delta^{n-1}$ can be written as:

$$\frac{d\phi(\mathbf{w})}{d\mathbf{w}} = \mathbf{J}(\phi(\mathbf{w})). \tag{6}$$

The matrix-valued function $\boldsymbol{J}$ will play an important role in our analysis of the gradient dynamics. We will therefore state some of its useful properties when restricted to $\mathring{\Delta}^{n-1}$ (See Appendix for the proof):

**Lemma 1.** *Let $\mathbf{z}$ be a vector in $\mathring{\Delta}^{n-1}$. The matrix $\boldsymbol{J}(\mathbf{z})$ satisfies the following:*

1. *$\boldsymbol{J}(\mathbf{z})$ is a symmetric positive semidefinite matrix.*

2. *The vector $\mathbb{1}_n$ is the eigenvector associated with the eigenvalue $0$, i.e. $\boldsymbol{J}(\mathbf{z})\mathbb{1}_n = 0$.*

3. *Let $\lambda_1(\mathbf{z}) \leq \cdots \leq \lambda_n(\mathbf{z})$ denote the eigenvalues of $\boldsymbol{J}(\mathbf{z})$ and let $\tilde{\mathbf{z}}$ be a vector whose entries are the entries of $\mathbf{z}$ sorted in ascending order $\tilde{\mathbf{z}}_i \leq \tilde{\mathbf{z}}_{i+1}$ for $i \in [n-1]$. The eigenvalues satisfy*

$$0 = \lambda_1(\mathbf{z}) < \tilde{\mathbf{z}}_1 \leq \lambda_2(\mathbf{z}) \leq \cdots \leq \lambda_n(\mathbf{z}) \leq \tilde{\mathbf{z}}_n < 1. \tag{7}$$

## 3 MAIN RESULTS

This section analyzes the dynamics of GD for our softmax attention model through two distinct yet complementary perspectives. First, we leverage standard regularity conditions on the loss to establish a linear convergence rate in the overparameterized regime. The analysis highlights optimization efficiency in this setting, but shows the rate worsens as attention vectors become sparser. Nevertheless, it fails to account for cases that are observed in practice and does not explain the mechanism underlying the poor performance of GD. To address these limitations, the second subsection explores gradient descent dynamics without assuming overparameterization. Due to the inherent complexities and the current lack of robust technical tools for a global analysis in this regime, we focus on a local analysis around global solutions and show that the softmax map can induce severe ill-conditioning, particularly when attention becomes sparse, which can lead to slower convergence.

Together, these analyses provide a nuanced understanding of GD dynamics in softmax attention models, highlighting both the strengths of overparameterized settings and the challenges faced in more realistic, constrained parameter regimes.

### 3.1 GLOBAL CONVERGENCE IN THE OVERPARAMETERIZED CASE

Here, we establish the global convergence of GD using a common strategy that relies on two main ingredients: the Lipschitz continuity of the gradient of the loss, and the Polyak-Łojasiewicz (PŁ) inequality (Polyak, 1963). These two conditions only need to hold along the trajectories, as referenced in Nguyen (2021), and Wu et al. (2023).

With $\boldsymbol{W}$ and $\boldsymbol{v}$ frozen, the GD update is given by:

$$\boldsymbol{p}_{t+1} = \boldsymbol{p}_t - \frac{\eta}{N} \sum_{i=1}^{N} \boldsymbol{K}_i^\top \boldsymbol{J}(\phi(\boldsymbol{K}_i \boldsymbol{p}_t)) \boldsymbol{X}_i \boldsymbol{v} \nabla \ell(f(\boldsymbol{X}_i), y_i). \tag{8}$$

Define

$$\mathbf{F}(\boldsymbol{p}_t) := \frac{1}{N} \begin{bmatrix} (\boldsymbol{K}_1^\top \boldsymbol{J}(\phi(\boldsymbol{K}_1 \boldsymbol{p}_t)) \boldsymbol{X}_1 \boldsymbol{v})^\top \\ \vdots \\ (\boldsymbol{K}_N^\top \boldsymbol{J}(\phi(\boldsymbol{K}_N \boldsymbol{p}_t)) \boldsymbol{X}_N \boldsymbol{v})^\top \end{bmatrix} \in \mathbb{R}^{N \times d} \tag{9}$$

We can rewrite the gradient update equation in the following simplified form

$$\boldsymbol{p}_{t+1} = \boldsymbol{p}_t + \eta \mathbf{F}(\boldsymbol{p}_t)^\top \mathbf{g}(\mathbf{p_t}), \tag{10}$$

where $\mathbf{g}(\mathbf{p_t})$ is the vector of gradients of $\ell$ with respect to the output of the model, i.e. its $i$-th component is $\mathbf{g_i}(\mathbf{p_t}) = \nabla \ell(f(\boldsymbol{X}_i), y_i)$. The following lemma, which will be used to provide a PŁ inequality in our convergence proof, now follows from the identity $\nabla \mathcal{L}(\boldsymbol{p}) = \mathbf{F}(\boldsymbol{p})^\top \mathbf{g}(\boldsymbol{p})$.

**Lemma 2.** *For all $\boldsymbol{p} \in \mathbb{R}^d$, the loss satisfies the pointwise inequality:*

$$\tfrac{1}{2} \|\nabla \mathcal{L}(\boldsymbol{p})\|_2^2 \geq 2m\, \sigma_{\min}^2(\mathbf{F}(\boldsymbol{p}))(\mathcal{L}(\boldsymbol{p}) - \mathcal{L}^*). \tag{11}$$

The next theorem provides sufficient conditions to guarantee the linear convergence of GD. The conditions establish a uniform lower bound for $\mu(\boldsymbol{p})$ which depends on the initialization of the weight vector $\boldsymbol{p}$. Additionally, they ensure that the solutions remain bounded, thereby guaranteeing the Lipschitz continuity of the gradients.

**Theorem 1.** *(Linear convergence). Assume $d \geq N$. There exist positive quantities $L', \mu, \eta, \gamma, M_p$ (defined in Appendix E) such that:*

*(i) **Linear convergence under spectral margin condition**. If $\sigma_{\min}(\boldsymbol{F}(\boldsymbol{p}_0)) > \gamma + \sqrt{\mu/(2m)}$, then Gradient Descent with stepsize $\eta \in (0, 2/L')$ satisfies, for all $t \geq 0$,*

$$\mathcal{L}(t) - \mathcal{L}^* \leq (1 - \eta\mu)^t (\mathcal{L}(0) - \mathcal{L}^*), \qquad \|\boldsymbol{p}_t\|_2 \leq M_p.$$

*(ii) **Almost-sure margin condition**. Suppose that each $\boldsymbol{X}_i \in \mathbb{R}^{n \times d}$ has full row rank and is drawn from a continuous distribution. Let $\boldsymbol{v} = \alpha \boldsymbol{u}$ with $\|\boldsymbol{u}\|_2 = 1$. There exists $A < \infty$ such that for all $\alpha \geq A$,*

$$\sigma_{\min}(\boldsymbol{F}(\boldsymbol{p}_0)) > \gamma + \sqrt{\mu/(2m)} \quad \text{almost surely.}$$

**Remark 1.** *If $d < N$, then $\sigma_{\min}(\boldsymbol{F}(\boldsymbol{p}_0)) = 0$ and the margin condition is vacuous.*

**Remark 2.** *Although the proof for linear convergence also holds for a non-constant $\boldsymbol{v}$, we fix it to isolate the effect of the softmax dynamics on GD. This simplification not only clarifies the comparison with the results in the next subsection but also shows that linear convergence is attainable without a trainable final linear layer, unlike in Wu et al. (2023).*

**Limitations of the overparameterized setting.** One significant drawback of this style of analysis, which relies on the PL inequality, is that it requires that the norm of $\boldsymbol{p}$ remain bounded along its trajectory. However, to learn sparse or approximately sparse attention maps, the norm of the weight vector $\boldsymbol{p}$ must approach infinity. As the bound $C_p$ increases to allow for that, the convergence rate described in the theorem worsens and eventually yields a vacuous bound. Moreover, the ability to analyze gradient descent behavior in contexts where learned attention probabilities are approximately sparse is critical. The primary objective of the attention mechanism is to focus on the most relevant parts of the input, resulting in attention probabilities that are typically far from uniform. This has been confirmed by prior empirical observations (Wu et al., 2023; Li et al., 2023a; Chen et al., 2021), and can also be clearly observed across layers and heads in the attention weights that we included in Appendix I for the Vision Transformer model (Dosovitskiy et al., 2020) used in our experiments.

### 3.2 LOCAL DYNAMICS OF SOFTMAX ATTENTION: A PARAMETERIZATION-AGNOSTIC ANALYSIS

In practice, modern attention models often operate in an *underparameterized* regime with embedding dimension $d$ much smaller than the number of training examples $N$ (i.e., $d \ll N$); see, e.g., Vaswani et al. (2017). Theoretical tools like the PŁ inequality, which are central to the overparameterized analysis in Section 3.1 and prior work Allen-Zhu et al. (2019); Du et al. (2019); Nguyen (2021); Bombari et al. (2022), typically do not apply in this setting.

Without these tools, we shift our focus to a tractable local analysis around the solutions of the optimization dynamics. Our analysis departs from a parameter-centric view and instead considers the dynamics directly in the space of the attention probabilities. This perspective allows us to build an intuitive link between the model's learned behavior—namely, the sparsity of its attention—and the local geometry of the optimization landscape.

We begin with an overparameterized simplified setting in which the dynamics decouple across training examples: each example's attention evolves independently on its own probability simplex. This decoupling makes transparent how attention sparsity/concentration governs the local conditioning and thus the local convergence rate along that simplex. We then turn to the general coupled case, where interactions between examples shape each simplex's dynamics; our local bounds apply to both over- and under-parameterized models and quantify the effect of this coupling.

**Warm-up (Overparameterized Decoupled case).** Assume *cross-sample decoupling* $\boldsymbol{K}_i \boldsymbol{K}_j^\top = 0$ for $i \neq j$. This decoupling can be realized in *overparameterized* regimes (e.g., sufficient width to orthogonalize features). Define $\tilde{\ell}_i(\boldsymbol{x}) := \ell(\boldsymbol{x}^\top \boldsymbol{X}_i \boldsymbol{v}, y_i)$. The GD step

$$\boldsymbol{p}_{t+1} = \boldsymbol{p}_t - \frac{\eta}{N} \sum_{j=1}^N \boldsymbol{K}_j^\top J(\phi(\boldsymbol{K}_j \boldsymbol{p}_t)) \nabla \tilde{\ell}_j(\phi(\boldsymbol{K}_j \boldsymbol{p}_t))$$

induces per-example attention dynamics

$$\boldsymbol{a}_{i,t+1} = \phi\Big(\boldsymbol{K}_i \boldsymbol{p}_t - \tfrac{\eta}{N} J(\boldsymbol{a}_{i,t}) \nabla \tilde{\ell}_i(\boldsymbol{a}_{i,t})\Big), \qquad \boldsymbol{a}_{i,t} := \phi(\boldsymbol{K}_i \boldsymbol{p}_t),$$

so each $\boldsymbol{a}_i$ evolves independently on $\Delta^{n-1}$. A first-order expansion on $T\Delta^{n-1}$ gives

$$\boldsymbol{a}_{i,t+1} = \boldsymbol{a}_{i,t} - \frac{\eta}{N} J(\boldsymbol{a}_{i,t})^2 \nabla \tilde{\ell}_i(\boldsymbol{a}_{i,t}) + R_{i,t}, \qquad \|R_{i,t}\| = O\big((\eta/N)^2\big). \tag{12}$$

**Proposition 1** (Decoupled linearization and conditioning). *Let $\boldsymbol{a}_i^\star \in \text{int}(\Delta^{n-1})$ be a stationary point of equation 12 and set $H_i := \nabla^2 \tilde{\ell}_i(\boldsymbol{a}_i^\star) \succeq 0$ with $\mu_i = \lambda_{\min}(H_i) > 0$ and $L_i = \lambda_{\max}(H_i) < \infty$. On $T\Delta^{n-1}$, writing $\boldsymbol{\zeta}_{i,t} = \boldsymbol{a}_{i,t} - \boldsymbol{a}_i^\star$,*

$$\boldsymbol{\zeta}_{i,t+1} = \Big(I - \tfrac{\eta}{N} J(\boldsymbol{a}_i^\star)^2 H_i\Big) \boldsymbol{\zeta}_{i,t},$$

*and the condition number $\kappa_i$ of $J(\boldsymbol{a}_i^\star)^2 H_i$ obeys*

$$\kappa_i \;\geq\; \frac{\mu_i}{L_i}\left(\frac{\max_j(\boldsymbol{a}_i^\star)_j}{\min_j(\boldsymbol{a}_i^\star)_j}\right)^2,$$

*so as attention becomes sparse ($\min_j(\boldsymbol{a}_i^\star)_j \downarrow 0$), the best linear rate $1 - \frac{1}{\kappa_i}$ approaches 1.*

**Remark 3.** *The decoupled model exposes the conditioning problem cleanly: attention sparsity directly worsens conditioning (via $(\min_j(\boldsymbol{a}_i^\star)_j)^{-2}$) and results from the spectral properties of the softmax Jacobian $J(\boldsymbol{a}_i^\star)$. Note that this does not contradict the results from the previous subsection. Theorem 1 bounds $\|p_t\|$ uniformly, which rules out the norm blow-up needed to reach sparse, ill-conditioned minima, so GD never enters the slow-convergence region and instead converges to minima associated with more uniform attention and thus a better condition number.*

**Remark 4.** *Overparameterization is invoked here* solely *to justify the decoupling assumption in this warm-up. Our subsequent local results do* not *assume decoupling and apply in both over- and under-parameterized settings where cross-sample coupling is present.*

**From decoupled to coupled dynamics.** The decoupled model highlights the core mechanism: sparsity in the attention vector directly worsens conditioning through the Jacobian. In the full model, however, training examples are coupled through shared parameters, and the resulting dynamics mix different simplices. To analyze this general case, we need a way to quantify how strongly attention concentrates on certain tokens and how balanced the mass is across them. This motivates the introduction of *attention separation* parameters, which translate the raw score geometry into explicit lower and upper bounds on important vs. unimportant attention weights. These quantities will allow us to extend the conditioning analysis from the independent setting to the realistic coupled one.

**Attention separation.** For each training example $i \in [N]$ with keys $\boldsymbol{K}_i \in \mathbb{R}^{n \times d}$, let $\boldsymbol{a}_i^\star := \phi(\boldsymbol{K}_i\boldsymbol{p}^\star) \in \Delta^{n-1}$ be the attention at a global minimum $\boldsymbol{p}^\star$. We partition token indices into an important set $S_i \subset [n]$ with $s_i := |S_i|$ and an unimportant set $S_i^c$. We quantify score separation via the following two nonnegative quantities

$$\Delta_i \;:=\; \min_{j \in S_i}\langle \boldsymbol{K}_i[j,:], \boldsymbol{p}^\star\rangle - \max_{\ell \notin S_i}\langle \boldsymbol{K}_i[\ell,:], \boldsymbol{p}^\star\rangle, \qquad \Gamma_i \;:=\; \max_{j \in S_i}\langle \boldsymbol{K}_i[j,:], \boldsymbol{p}^\star\rangle - \min_{j \in S_i}\langle \boldsymbol{K}_i[j,:], \boldsymbol{p}^\star\rangle.$$

Define the global important/unimportant attention extremes

$$a_I \;:=\; \min_i \min_{j \in S_i}(\boldsymbol{a}_i^\star)_j, \qquad a_U \;:=\; \max_i \max_{\ell \notin S_i}(\boldsymbol{a}_i^\star)_\ell.$$

**Lemma 3** (Global attention bounds). *Let $\Delta_{\min} := \min_i \Delta_i$, $\Gamma_{\max} := \max_i \Gamma_i$, $s_{\min} := \min_i s_i$, and $s_{\max} := \max_i s_i$. Then*

$$a_I \;\geq\; \frac{1}{1 + (s_{\max} - 1)e^{\Gamma_{\max}} + (n - s_{\min})e^{-\Delta_{\min}}}, \tag{13}$$

$$a_U \;\leq\; \frac{1}{1 + s_{\min}e^{\Delta_{\min}}}, \tag{14}$$

*and when all important tokens have uniform attention ($\Gamma_i = 0$ for all $i$), the sharper bound $a_U \leq \dfrac{e^{-\Delta_{\min}}}{s_{\min} + (n - s_{\max})e^{-\Delta_{\min}}}$ holds.*

**Remark 5.** *The margin $\Delta_i$ measures the score gap between important and unimportant tokens via alignment with $\boldsymbol{p}^*$; softmax then amplifies this gap into a probabilistic one scaling with $e^{\Delta_{\min}}$. Within the important set $S_i$, the smallest attention weight also reflects score imbalance: a large spread $\Gamma_i$ lets one key dominate, lowering the minimum through the $(s_i - 1)e^{\Gamma_i}$ factor in equation 13. When $\Gamma_i$ is small (scores nearly equal), the mass is split almost uniformly and tails vanish exponentially; as $\Delta_i \to \infty$, $(\boldsymbol{a}_i^\star)_j \to 1/s_i$ for $j \in S_i$ and $(\boldsymbol{a}_i^\star)_\ell \to 0$ otherwise. Thus, probability-level quantities in our conditioning analysis can be replaced by the more interpretable parameters $(\Delta_i, \Gamma_i, s_i)$.*

Next, we introduce the structural conditions used later to establish our results.

**Assumption B.** *Fix an orthogonal decomposition $\mathbb{R}^d = I \oplus U$ with projectors $\boldsymbol{P}_I, \boldsymbol{P}_U$. For each $i \in [N]$:*

*(B1) **Approximate Subspace geometry:** Let $\boldsymbol{K}_{i,S_i}$ and $\boldsymbol{K}_{i,S_i^c}$ collect the rows of $\boldsymbol{K}_i$ indexed by $S_i$ and $S_i^c$ respectively. Define*

$$\Sigma_U := \frac{1}{N}\sum_{i=1}^N (\boldsymbol{K}_{i,S_i^c}\boldsymbol{P}_U)^\top(\boldsymbol{K}_{i,S_i^c}\boldsymbol{P}_U).$$

*There exists $\rho_U \in [0,1)$ such that for all unit $x_U \in U$,*
$$\|\boldsymbol{K}_{i,S_i}x_U\|_2 \ \leq\ \rho_U\,\|\boldsymbol{K}_{i,S_i^c}x_U\|_2.$$
*(The case $\rho_U = 0$ recovers exact alignment of important keys with $I$.)*

*(B2) **Value–compatibility margin:** Let $\boldsymbol{u}_i := \boldsymbol{X}_i v$ and $\mu_i := \boldsymbol{a}_i^{\star\top}\boldsymbol{u}_i$. Assume there exists $\delta_i \geq 0$ such that for all $j \in S_i$,*
$$|(\boldsymbol{u}_i)_j - \mu_i| \ \geq\ \delta_i.$$

**Remark 6.** *(B1) states that important keys of $\boldsymbol{K}_i$ lie mostly in a common signal subspace $I$ and unimportant ones in $U$. (B2) states that, on tokens receiving large attention, the (projected) value magnitudes are uniformly separated from the global average $\mu_i = \sum_j(\boldsymbol{a}_i^\star)_j(\boldsymbol{u}_i)_j$. In standard transformer training, heads concentrate on tokens whose key/query scores are large; the value pathway then learns to place signal on those same coordinates (e.g., for copying, routing, or aggregation). These conditions strengthen the baseline assumptions in §2 without conflict.*

**Induced attention dynamics.** At a global minimum, the dynamics can be linearized either in the attention variables $\boldsymbol{a}_i$ or directly in the parameters $\boldsymbol{p}$. Linearizing in the simplex produces the block Jacobian
$$\mathcal{J}(\boldsymbol{a}^\star) = \big[A_{ik}\big]_{i,k=1}^N, \qquad A_{ik} = J(\boldsymbol{a}_i^\star)\boldsymbol{K}_i\boldsymbol{K}_k^\top J(\boldsymbol{a}_k^\star)H_k,$$
which makes the cross-example coupling explicit. However, $\mathcal{J}(\boldsymbol{a}^\star)$ is not symmetric, so its eigenvalues are difficult to control directly. Fortunately, $\mathcal{J}(\boldsymbol{a}^\star)$ admits a factorization $\mathcal{J} = BC$ whose nonzero eigenvalues coincide with those of the symmetric product $CB := \boldsymbol{H}_p$. This symmetric matrix is precisely what arises from the *parameter-space linearization*:
$$\boldsymbol{e}_{t+1} \ =\ \Big(I - \tfrac{\eta}{N}\boldsymbol{H}_p\Big)\boldsymbol{e}_t, \qquad \boldsymbol{e}_t = \boldsymbol{p}_t - \boldsymbol{p}^\star,$$
with
$$\boldsymbol{H}_p \ =\ \sum_{i=1}^N \boldsymbol{K}_i^\top J(\boldsymbol{a}_i^\star)\,H_i\,J(\boldsymbol{a}_i^\star)\boldsymbol{K}_i, \qquad H_i = \ell''(s_i^\star;y_i)\,\boldsymbol{u}_i\boldsymbol{u}_i^\top, \quad \boldsymbol{u}_i = \boldsymbol{X}_i v.$$

Thus, while the simplex perspective highlights how sparsity affects conditioning through the Jacobian, it is the **symmetric operator $\boldsymbol{H}_p$ from the parameter-space linearization** that fully governs local convergence. Its spectrum determines both stability and the worst-case rate.

**Theorem 2** (Lower bound on the condition number of $\boldsymbol{H}_p$). *Under Assumptions B, we have*
$$\kappa(\boldsymbol{H}_p) \ \geq\ \Omega\Big(\frac{a_I^2}{\rho_U^2 + (n - s_{\min})\,a_U^2}\Big).$$

*Here, the $\Omega(\cdot)$ notation hides a multiplicative constant that depends only on fixed quantities such as the strong convexity and Lipschitz smoothness constants ($m$ and $L$), the norms of the model parameters and data ($\boldsymbol{v}$, $W$, $\boldsymbol{X}_i$), as well as the sparsity levels $s_i$ of the attention vectors.*

**Remark 7.** *Under perfect token alignment ($\rho_U = 0$), the condition number grows with $\exp(2\Delta_{min})$ which highlights the blow up under a very large margin (tokens with very small attention).*

**Corollary 1** (Worst-Case Local Convergence Rate). *Let the assumptions of Theorem 2 hold. Let $\boldsymbol{p}^*$ be a global minimum and $\boldsymbol{v}_{\min}$ an eigenvector of $\lambda_{\min}(\boldsymbol{H}_p)$. Define the **initial alignment** of the error vector $\boldsymbol{e}_0 = \boldsymbol{p}_0 - \boldsymbol{p}^*$ with the slowest convergence direction as: $\cos^2(\theta_0) = (\boldsymbol{e}_0^\top\boldsymbol{v}_{\min})^2/\|\boldsymbol{e}_0\|_2^2$, where $\boldsymbol{p}_0$ is the parameter vector upon entering the local neighborhood.*

*With a step size $\eta$ satisfying the stability condition $\eta \leq 2/\lambda_{\max}$, the component of the error in the direction of $\boldsymbol{v}_{\min}$ evolves as:*
$$(\boldsymbol{e}_t^\top\boldsymbol{v}_{\min}) = (1 - \eta\lambda_{\min})^t(\boldsymbol{e}_0^\top\boldsymbol{v}_{\min}) \tag{15}$$
*In particular, for $\eta = 1/\lambda_{\max}$,*
$$\|\boldsymbol{e}_t\|_2^2 \geq \Big(1 - \frac{1}{\kappa}\Big)^{2t}\|\boldsymbol{e}_0\|_2^2\cos^2(\theta_0) \tag{16}$$

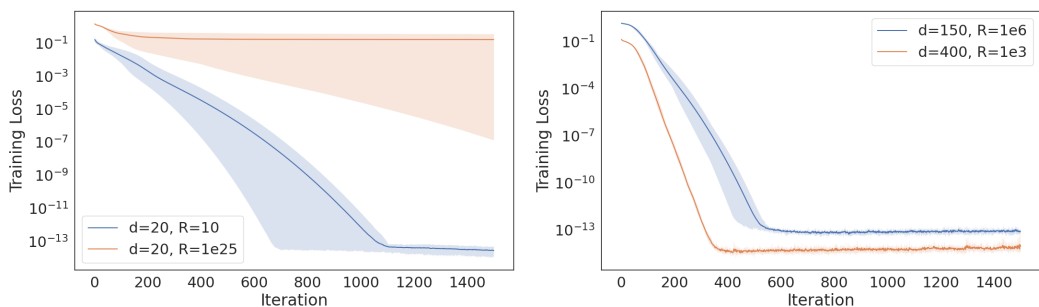

Figure 1: Training loss on synthetic data: GD slows down under sparse attention in the underparameterized regime (left) but not when overparameterized (right).

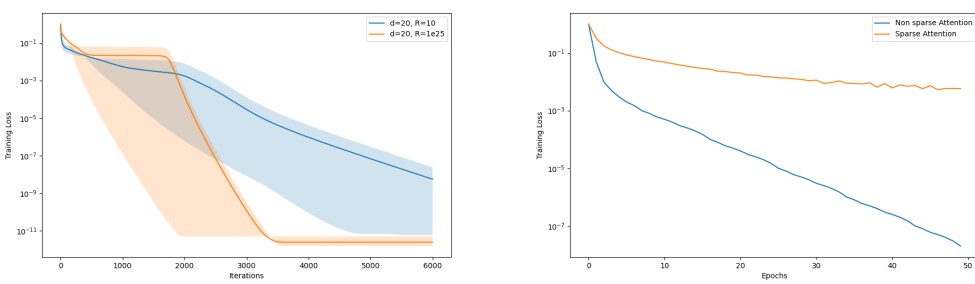

(a) Training loss under linear attention (no softmax).     (b) Tiny ViT training loss (avg over three runs).

Figure 2: Comparison of training losses across (a) linear attention and (b) softmax-based Tiny ViT, illustrating slower GD convergence with sparse softmax attention.

**Interpretation.** Corollary 1 makes the consequence of our condition number bound explicit. It shows that if the initial error $e_0$ has any non-zero alignment with the slow eigenspace (i.e., $\cos^2(\theta_0) > 0$), which is almost certain for any realistic optimization trajectory, the overall error is lower-bounded by a term that decays extremely slowly. As the attention probabilities become sparse, Theorem 2 implies that $\kappa \to \infty$, causing the convergence factor $(1 - 1/\kappa)$ to approach 1. This leads to the stagnation observed in practice.

## 4 EXPERIMENTAL RESULTS

**Simplified One-layer Softmax Attention Model.** First, we consider the training of the model defined in equation 2 using a squared loss. We set $W = I_d$, use full-batch GD on $p, v$, initialize $p, v \sim \mathcal{N}(0, 1/d)$, and draw $X_i$ i.i.d. standard Gaussian with $n = d$ on $N = 100$ samples. We vary width $d \in \{20, 150, 400\}$ and control attention sparsity via the ratio $R$ of largest to smallest attention probability at the end of training (higher $R$ = presence of tokens with very small attention), tuned through teacher variance $\sigma_p \in \{0.1, 1\}$ yielding $R_{\text{teacher}} \approx 10$ and $10^{25}$ (with $\sigma_v^2 = 1$). As shown in Fig. 1, GD in the underparameterized setting slows or stalls for large $R$ (stationary points near the simplex boundary), but converges when $R$ is small, consistent with Theorem 2. In the overparameterized setting, GD exhibits linear convergence and reaches solutions bounded away from the boundary as predicted by Theorem 1, with $R \ll R_{\text{teacher}}$. These findings remain unchanged when $W$ is also trained; see Appendix J.1 for the corresponding figure.

**Comparison with Linear Attention.** Our theory predicts that removing softmax should eliminate the slowdown. To test this, we repeated the experiment of Figure 1 with a linear attention mechanism, using the same teacher datasets in both sparse ($R \approx 10^{25}$) and non-sparse ($R = 10$) regimes. Figure 2a confirms that without softmax, Gradient Descent converges quickly even under sparse attention, validating our conclusion that the slowdown is induced by the softmax nonlinearity.

**Experiment with Vision Transformer (ViT) on MNIST.** We extend our analysis to more realistic scenarios by training a Vision Transformer (ViT) with 6 layers and 4 heads on the MNIST dataset

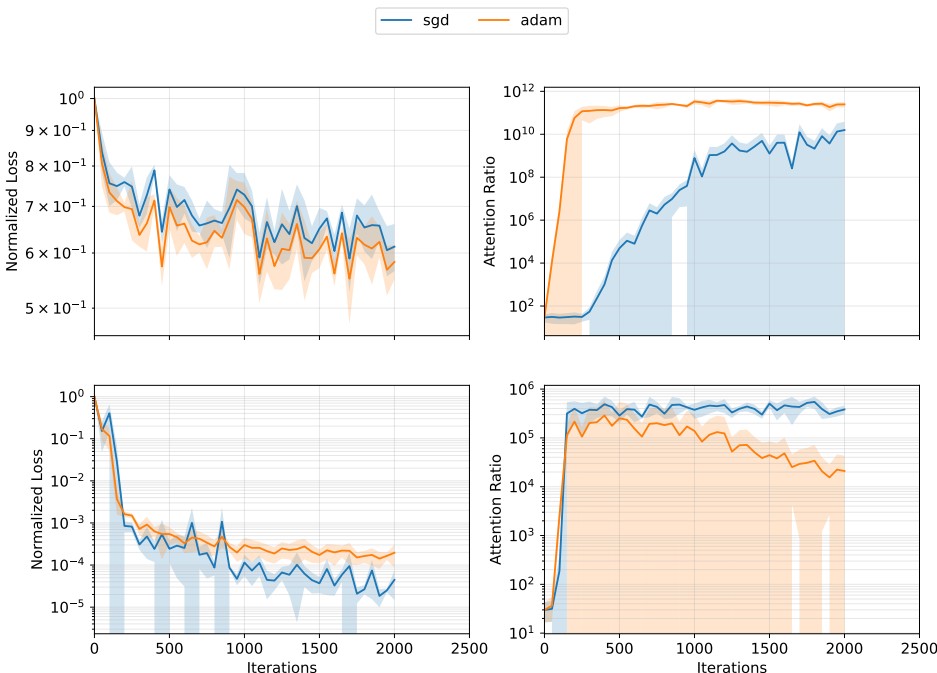

Figure 3: Training loss (left) and attention ratio (right) for two WikiText-2 tasks. Top row: WikiText-2 next-token prediction. Borrom row: WikiText-2 bag-of-words classification.

(Figure 2b). We choose a patch size of 4, the embedding dimension that we consider is 64 and the MLP dimension is 256. The labels are once again generated using teacher models that enforce different sparsity levels of the attention scores. The ratio $R$ across layers and heads for the sparse attention setting (orange curve) is $R \approx 10^7$, whereas for the non sparse or less sparse version (blue curve), the average ratio is $R \approx 20$. The model is trained using Stochastic Gradient Descent (SGD) with a constant step size $\eta = 0.01$ that was chosen using a grid search over the step sizes $[0.001, 0.01, 0.1]$. The experiment shows that the conclusions obtained from the theoretical analysis of the simplified model also hold for more practical architectures such as the Vision Transformer : (S)GD suffers from slow convergence as the attention probabilities become sparser.

**Experiment on WikiText-2.** We trained a two-layer transformer with four heads and hidden size 128 on WikiText-2 using two contrasting tasks: (*i*) next-token prediction, which induces sparse attention distributions, and (*ii*) a bag-of-words classification task (predicting whether a sequence contains the word "the"), which yields more uniform attention. We compared SGD (with momentum) and Adam under tuned learning rates, and tracked both training loss and the attention ratio of the CLS token. Results (Figure3) show that SGD stagnates in the sparse-attention task, consistent with our condition number analysis, while Adam continues to converge. In the uniform-attention task, Adam no longer outperforms SGD. These findings support our theoretical results connecting the poor performance of GD to the sparsity of the softmax attention.

## 5 CONCLUSION

We analyzed the optimization dynamics of gradient descent in attention-based models, isolating the role of the softmax nonlinearity. In overparameterized regimes, we showed that linear convergence is possible under standard PL and smoothness conditions, but this analysis breaks down in practice. In the underparameterized regime, our local results reveal that convergence degrades as attention becomes sparse, since softmax drives the optimization landscape toward ill-conditioning. While our study relies on simplifying assumptions and local analyses, it identifies a fundamental mechanism behind the failure of gradient descent in transformers and motivates the design of adaptive or preconditioned algorithms that can mitigate these effects in realistic settings.

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

## A    USE OF LARGE LANGUAGE MODELS

We used a large language model (LLM) as a general-purpose writing and coding assistant during the preparation of this paper. Specifically, the LLM was employed to: (i) generate initial code skeletons and boilerplate for experiments, (ii) assist with brainstorming and exploring alternative framings of technical ideas, and (iii) help polish the exposition by suggesting rephrasings and stylistic improvements. All theoretical results, proofs, and core experimental designs were conceived and developed by the authors.

## B    APPENDIX

## C    RELATED WORK

**Dynamics of Attention (Representation Learning).** A first line of work analyzes how attention layers acquire structure during training. Li et al. (2023c) study a single-layer transformer and show the emergence of semantic structure; a single head can learn to focus on salient tokens (Snell et al., 2021); and simplified ViTs can recover spatial structure without locality priors (Jelassi et al., 2022). Recent results further characterize training trajectories and phase transitions: in shallow settings, attention evolves from uniform to sparse via a scan-and-snap dynamic (Tian et al., 2023; Tian et al.), and incremental rank growth appears under diagonal attention with small initialization (Abbe et al., 2023).

**Dynamics of Attention (Convergence & Implicit Bias).** A second line focuses on optimization dynamics and implicit bias. For simplified attention models trained by gradient descent, Tarzanagh et al. (2023a;b) show SVM-type implicit bias along regularization paths. Finite-time guarantees have been obtained for adaptive step-size variants such as normalized GD and the Polyak step size (Vasudeva et al.). In logistic-loss settings, Deora et al. establish loss convergence under suitable initializations with polylogarithmically many heads. With pretraining-style initialization and hinge loss, Liu et al. (2023) analyze sample complexity under a relevant/irrelevant token model and show that attention weights concentrate only at a sublinear rate during training. Overall, much of this line either assumes overparameterization or imposes specialized data models to obtain sharp statements.

Closest to our results, Wu et al. (2023) prove linear convergence for a one-layer self-attention model under $\mathcal{O}(N)$ overparameterization, with the guarantee relying on the final linear output layer; the underparameterized regime is not treated. Our analysis targets precisely this regime. We isolate how the transition toward sparse attention collapses the smallest eigenvalues of the local curvature, which in turn predicts slow SGD. At the same time, we provide sufficient conditions under which GD still achieves a linear rate (Theorem 1), and—crucially—our argument derives the rate from the weights inside the softmax layer itself rather than from the output head. This view complements scan-and-snap dynamics by tying sparsification directly to conditioning and convergence behavior.

**Failure of SGD on Transformers.** Several works have explored why SGD performs poorly on transformers. Zhang et al. (2020) hypothesized that it is caused by the heavy-tailed stochastic noise in language tasks. However, Kunstner et al. (2023) found that the SGD fails even in the full-batch case. Moreover, Zhang et al. (2024a) showed that Vision Transformers trained with SGD on ImageNet suffer from the same problems. Therefore, the stochasticity of the gradients and the data modality cannot explain the failure of SGD on transformers. Pan & Li (2023) introduce the notion of directional sharpness and argue that SGD has a high directional sharpness, which they claim is correlated with low performance of optimization algorithms. Jiang et al. (2023) propose a new notion of condition number and show that it is large on the path taken by SGD. Tomihari & Sato (2025) argue that gradient heterogeneity explains why SGD performs poorly in transformers. Yadav et al. (2023) hypothesize that the poor performance of SGD is due to the heavy-tailed class imbalance. They empirically show that other architectures such as CNNs suffer from the same issue under heavy-tailed class imbalance. Zhao et al. empirically demonstrate that adaptive optimization is primarily beneficial for the last layer and layer normalization parameters in transformers. Our analysis provides a complementary theoretical explanation: these components involve softmax operations (output layer) or interact closely with attention distributions (layer norms), making them susceptible to the ill-conditioning we characterize.

**Transformer Hessian Analysis.** Zhang et al. (2024a) compute the spectrum of the Hessian and show that, across blocks, it varies significantly for transformers, unlike other architectures like CNNs and MLPs. Their theoretical results are however limited to quadratic models. Ormaniec et al. (2025) derive an exact, block-structured Hessian for self-attention and show that curvature is governed by first–third centered moments of the attention distribution; as attention rows approach one-hot, the query–key blocks shrink, accentuating inter-block anisotropy and helping to explain the usefulness of adaptive methods. This insight is leveraged by Zhang et al. (2024b) to design a more memory-efficient variant of Adam. Our analysis is complementary: rather than global Hessian structure, we study the local convergence operator proving that attention sparsity and key-vector alignment drive spectral collapse, thereby quantifying SGD slow-down.

**Rank Collapse in Transformers.** A related line of work studies *rank collapse*, whereby token representations become increasingly correlated with depth. Dong et al. (2021) prove that pure self-attention (without skip connections or MLPs) causes outputs to converge doubly exponentially to a rank-1 matrix, driving attention toward uniformity. Noci et al. (2022) show that this high correlation among tokens causes gradients of queries and keys to vanish at initialization, hindering training. Our analysis addresses a complementary regime: rather than uniform attention arising from rank collapse, we study *sparse* attention where mass concentrates on few tokens. In both cases, the attention distribution deviates from a well-conditioned intermediate regime, but through opposite mechanisms—rank collapse pushes attention toward uniformity while task-driven learning often pushes it toward sparsity. Together, these perspectives provide a more complete picture of attention-related optimization pathologies in transformers.

**Softmax-induced Optimization Issues.** Hoffmann et al. (2024) study "Eureka-moments" in transformers and trace long plateaus in training loss to the Softmax in self-attention: for certain ill-distributed attention patterns they observe that gradients for the query and key weights become much smaller than those for the value weights, and show that modifying the Softmax restores healthy gradients and removes the plateau. Our work is complementary: in a simplified one-layer attention model with a single tunable token we derive the exact linearization of gradient descent in the attention simplex and show that sparse attention makes the Softmax Jacobian ill-conditioned, which in turn causes a spectral collapse of the local GD operator.

## D  PROOF OF LEMMA 1

**Lemma 1.** *Let $\mathbf{z}$ be a vector in $\mathring{\Delta}^{n-1}$. The matrix $\boldsymbol{J}(\mathbf{z})$ satisfies the following:*

1. *$\boldsymbol{J}(\mathbf{z})$ is a symmetric positive semidefinite matrix.*

2. *The vector $\mathbb{1}_n$ is the eigenvector associated with the eigenvalue 0, i.e. $\boldsymbol{J}(\mathbf{z})\mathbb{1}_n = 0$.*

3. *Let $\lambda_1(\mathbf{z}) \leq \cdots \leq \lambda_n(\mathbf{z})$ denote the eigenvalues of $\boldsymbol{J}(\mathbf{z})$ and let $\tilde{\mathbf{z}}$ be a vector whose entries are the entries of $\mathbf{z}$ sorted in ascending order $\tilde{\mathbf{z}}_i \leq \tilde{\mathbf{z}}_{i+1}$ for $i \in [n-1]$. The eigenvalues satisfy*

$$0 = \lambda_1(\mathbf{z}) < \tilde{\mathbf{z}}_1 \leq \lambda_2(\mathbf{z}) \leq \cdots \leq \lambda_n(\mathbf{z}) \leq \tilde{\mathbf{z}}_n < 1. \tag{7}$$

*Proof.* Let $\boldsymbol{z} \in \Delta^{n-1}$. Recall that $\boldsymbol{J}(\boldsymbol{z}) = \mathrm{diag}(\boldsymbol{z}) - \boldsymbol{z}\boldsymbol{z}^\top = \mathrm{diag}(\boldsymbol{z})(I_n - \mathbb{1}_n\boldsymbol{z}^\top)$.

1. We have $\boldsymbol{J}(\boldsymbol{z})^\top = (\mathrm{diag}(\boldsymbol{z}) - \boldsymbol{z}\boldsymbol{z}^\top)^\top = \boldsymbol{J}(\boldsymbol{z})$, therefore $\boldsymbol{J}(\boldsymbol{z})$ is symmetric. Moreover, $\boldsymbol{J}(\boldsymbol{z})$ is the product of a diagonal positive semidefinite matrix and a projection, therefore it is positive semidefinite.

2. $\boldsymbol{J}(\boldsymbol{z})\mathbb{1}_n = \mathrm{diag}(\boldsymbol{z})\mathbb{1}_n - \boldsymbol{z}\boldsymbol{z}^\top\mathbb{1}_n\boldsymbol{z} - \boldsymbol{z} = 0$.

3. To prove the inequality on the eigenvalues, consider Corollary 4.3.5 in Horn & Johnson (1985) which we restate here:

**Corollary 2.** *Horn & Johnson (1985) Let $A, B \in M_n$ be Hermitian. Suppose that $B$ is singular and rank $B = r$. Then*

$$\lambda_i(A + B) \leq \lambda_{i+r}(A), \quad i = 1, \ldots, n - r \tag{17}$$

By setting $A = \boldsymbol{J}(\boldsymbol{z})$ and $B = \boldsymbol{z}\boldsymbol{z}^\top$, we have that both matrices are symmetric, $\mathrm{rank}(B) = 1$, and $A + B = \mathrm{diag}(\boldsymbol{z})$. Therefore

$$\lambda_i(\mathrm{diag}(\boldsymbol{z})) \leq \lambda_{i+1}(\boldsymbol{J}(\boldsymbol{z})) \tag{18}$$

Since the eigenvalues of $\mathrm{diag}(z)$ are $z_1, \dots, z_n$, we just need to order them from smallest to largest and we get

$$\tilde{\mathbf{z}}_1 \leq \lambda_2(\mathbf{z}) \leq \cdots \leq \lambda_n(\mathbf{z}) \leq \tilde{\mathbf{z}}_n \tag{19}$$

Finally, since $\boldsymbol{z}$ is in the interior, we know that $\tilde{\mathbf{z}}_1 > 0$ and $\tilde{\mathbf{z}}_n < 1$, which concludes our proof. $\square$

# E  PROOF OF THEOREM 1

**Theorem 1.** *(**Linear convergence**).* *Assume $d \geq N$. There exist positive quantities $L', \mu, \eta, \gamma, M_p$ (defined in Appendix E) such that:*

*(i)* ***Linear convergence under spectral margin condition**. If $\sigma_{\min}(\boldsymbol{F}(\boldsymbol{p}_0)) > \gamma + \sqrt{\mu/(2m)}$, then Gradient Descent with stepsize $\eta \in (0, 2/L')$ satisfies, for all $t \geq 0$,*

$$\mathcal{L}(t) - \mathcal{L}^* \leq (1 - \eta\mu)^t (\mathcal{L}(0) - \mathcal{L}^*), \qquad \|\boldsymbol{p}_t\|_2 \leq M_p.$$

*(ii)* ***Almost-sure margin condition**. Suppose that each $\boldsymbol{X}_i \in \mathbb{R}^{n \times d}$ has full row rank and is drawn from a continuous distribution. Let $\boldsymbol{v} = \alpha \, \boldsymbol{u}$ with $\|\boldsymbol{u}\|_2 = 1$. There exists $A < \infty$ such that for all $\alpha \geq A$,*

$$\sigma_{\min}(\boldsymbol{F}(\boldsymbol{p}_0)) > \gamma + \sqrt{\mu/(2m)} \quad \text{almost surely.}$$

First, we prove the following lemma:

**Lemma 4.** *Suppose $d \geq N$, and let $c_1$ and $c_2$ be some positive constants such that $c_1 < 1$ and $c_2 < \min(2, 1/c_1)$. Define the following quantities:*

$$C_\sigma = \max_{1 \leq i \leq N} \sigma_{max}(\boldsymbol{X}_i), \quad L' = \sigma_{max}^2(\boldsymbol{W}) C_\sigma^3 L \|\boldsymbol{v}\| \left( 2C_\sigma \|\boldsymbol{v}\| + 3\sqrt{\frac{2(\mathcal{L}_0 - \mathcal{L}^*)}{m}} \right)$$

$$\mu = \frac{c_1}{L'}, \quad \eta = \frac{c_2}{L'}$$

$$C_p = \sqrt{\frac{2(\mathcal{L}_0 - \mathcal{L}^*)}{mN}} \frac{\sigma_{max}(\boldsymbol{W}) C_\sigma^2 \|\boldsymbol{v}\| c_2 L}{L'(1 - \sqrt{1 - c_1 c_2})}, \quad M_p = \|\boldsymbol{p}_0\| + C_p$$

$$\gamma = \frac{1}{\sqrt{N}} (\min(2, 3C_\sigma \sigma_{max}(\boldsymbol{W}) C_p) C_\sigma^2 \sigma_{max}(\boldsymbol{W}) \|\boldsymbol{v}\|_2)$$

*If $\sigma_{min}(\mathbf{F}(\boldsymbol{p}_0)) > \gamma + \sqrt{\frac{\mu}{2m}}$, then gradient descent on the function $\mathcal{L}$ with step size $\eta$ converges at a linear rate:*

$$\mathcal{L}(t) - \mathcal{L}^* \leq (1 - \eta\mu)^t (\mathcal{L}(0) - \mathcal{L}^*), \quad \text{for all } t \geq 0. \tag{20}$$

*Moreover, the weight vector $\boldsymbol{p}$ remains bounded throughout the trajectory*

$$\|\boldsymbol{p}_t\|_2 \leq M_p, \quad \text{for all } t \geq 0.$$

*Proof.* We will show by strong induction that for every iteration $t \geq 0$

$$\begin{cases} \|\boldsymbol{p}_t\|_2 \leq M_p \\ \mathcal{L}(t) - \mathcal{L}^* \leq (1 - \eta\mu)^t (\mathcal{L}(0) - \mathcal{L}^*), & \text{for all } t \geq 0 \end{cases} \tag{21}$$

We start by deriving an upper bound on the gradient of $\mathcal{L}$:

$$\|\nabla_{\boldsymbol{p}} \mathcal{L}_t\|_2 = \frac{1}{N} \| \sum_{i=1}^N \boldsymbol{K}_i^\top \boldsymbol{J}(\phi(K_i \boldsymbol{p}_t)) \boldsymbol{X}_i \boldsymbol{v} \nabla \ell(\phi(\boldsymbol{K}_i \boldsymbol{p}_t)^\top \boldsymbol{X}_i \boldsymbol{v}, y_i) \|_2$$

$$\leq \frac{1}{N} \sum_{i=1}^N \| \boldsymbol{K}_i^\top \boldsymbol{J}(\phi(K_i \boldsymbol{p}_t)) \boldsymbol{X}_i \boldsymbol{v}_t \nabla \ell(\phi(\boldsymbol{K}_i \boldsymbol{p}_t)^\top \boldsymbol{X}_i \boldsymbol{v}_t, y_i) \|_2$$

$$\leq \frac{1}{N} \sum_{i=1}^N |\nabla \ell(\phi(\boldsymbol{K}_i \boldsymbol{p}_t)^\top \boldsymbol{X}_i \boldsymbol{v}, y_i)| \sigma_{max}(\boldsymbol{X}_i) \sigma_{max}(\boldsymbol{K}_i) \sigma_{max}(\boldsymbol{J}(\phi(\boldsymbol{K}_i \boldsymbol{p}_t))) \|\boldsymbol{v}\|_2$$

$$\leq \sqrt{\frac{2(\mathcal{L}_t - \mathcal{L}^*)}{mN}} L C_\sigma^2 \sigma_{max}(\boldsymbol{W}) \|\boldsymbol{v}\|_2 \tag{22}$$

where the second inequality is a result of the triangle inequality and the third inequality uses the sub-multiplicativity of the spectral norm. To obtain the last inequality, we have used the fact that $\sigma_{max}(\boldsymbol{J}) \leq 1$ as a consequence of Lemma 1.

Now suppose that the inductive hypothesis holds for all $s \in [t]$. We will use it to bound the deviations of the weights from their value at initialization:

$$
\begin{aligned}
\|\boldsymbol{p}_{t+1} - \boldsymbol{p}_0\|_2 &= \|\sum_{s=0}^{t} (\boldsymbol{p}_{s+1} - \boldsymbol{p}_s)\|_2 \\
&\leq \sum_{s=0}^{t} \|\boldsymbol{p}_{s+1} - \boldsymbol{p}_s\|_2 \\
&= \eta \sum_{s=0}^{t} \|\nabla_{\boldsymbol{p}} \mathcal{L}_s\|_2 \\
&\leq \eta \sum_{s=0}^{t} \sqrt{\frac{2(\mathcal{L}_s - \mathcal{L}^*)}{mN}} L C_\sigma^2 \sigma_{max}(\boldsymbol{W}) \|\boldsymbol{v}\|_2 \\
&\leq L\eta C_\sigma^2 \sigma_{max}(\boldsymbol{W}) \|\boldsymbol{v}\|_2 \sqrt{\frac{2}{mN}} \sum_{s=0}^{t} \sqrt{(1 - \eta\mu)^s (\mathcal{L}_0 - \mathcal{L}^*)} \\
&\leq L\eta C_\sigma^2 \sigma_{max}(\boldsymbol{W}) \|\boldsymbol{v}\|_2 \sqrt{\frac{2(\mathcal{L}_0 - \mathcal{L}^*)}{mN}} \sum_{s=0}^{+\infty} \sqrt{(1 - \eta\mu)^s} \\
&= \sqrt{\frac{2(\mathcal{L}_0 - \mathcal{L}^*)}{mN}} \frac{L\eta C_\sigma^2 \sigma_{max}(\boldsymbol{W}) \|\boldsymbol{v}\|_2}{1 - \sqrt{1 - \eta\mu}} = C_p
\end{aligned}
\tag{23}
$$

Therefore, we have:

$$
\|\boldsymbol{p}_{t+1}\|_2 \leq \|\boldsymbol{p}_0\|_2 + C_p = M_p
\tag{24}
$$

Our next step consists in showing local smoothness on the interval $[\boldsymbol{p}_t, \boldsymbol{p}_{t+1}]$. We define $\boldsymbol{p}_{t+\tau} = \boldsymbol{p}_t + \tau(\theta_{t+1} - \boldsymbol{p}_t), \tau \in [0, 1]$

$$
\begin{aligned}
\|\nabla_{\boldsymbol{p}} \mathcal{L}(\boldsymbol{p}_{t+\tau}) - \nabla_{\boldsymbol{p}} \mathcal{L}(\boldsymbol{p}_t)\|_2 &= \frac{1}{N} \left\| \sum_{i=1}^{N} \boldsymbol{K}_i^\top \left( \boldsymbol{J}(\phi(\boldsymbol{K}_i \boldsymbol{p}_{t+\tau})) \boldsymbol{X}_i \boldsymbol{v} \boldsymbol{g}_i(\boldsymbol{p}_{t+\tau}) - \boldsymbol{J}(\phi(\boldsymbol{K}_i \boldsymbol{p}_t)) \boldsymbol{X}_i \boldsymbol{v} \boldsymbol{g}_i(\boldsymbol{p}_t) \right) \right\|_2 \\
&\leq \frac{1}{N} \sum_{i=1}^{N} \left\| \boldsymbol{K}_i^\top \left( \boldsymbol{J}(\phi(\boldsymbol{K}_i \boldsymbol{p}_{t+\tau})) \boldsymbol{X}_i \boldsymbol{v} \boldsymbol{g}_i(\boldsymbol{p}_{t+\tau}) - \boldsymbol{J}(\phi(\boldsymbol{K}_i \boldsymbol{p}_t)) \boldsymbol{X}_i \boldsymbol{v} \boldsymbol{g}_i(\boldsymbol{p}_t) \right) \right\|_2 \\
&\leq \frac{1}{N} \sum_{i=1}^{N} \|\boldsymbol{K}_i\|_2 \|\boldsymbol{J}(\phi(\boldsymbol{K}_i \boldsymbol{p}_{t+\tau})) \boldsymbol{X}_i \boldsymbol{v} \boldsymbol{g}_i(\boldsymbol{p}_{t+\tau}) - \boldsymbol{J}(\phi(\boldsymbol{K}_i \boldsymbol{p}_t)) \boldsymbol{X}_i \boldsymbol{v} \boldsymbol{g}_i(\boldsymbol{p}_t)\|_2 \\
&\leq \frac{1}{N} C_\sigma \sigma_{max}(\boldsymbol{W}) \sum_{i=1}^{N} \|\boldsymbol{J}(\phi(\boldsymbol{K}_i \boldsymbol{p}_{t+\tau})) \boldsymbol{X}_i \boldsymbol{v} \boldsymbol{g}_i(\boldsymbol{p}_{t+\tau}) - \boldsymbol{J}(\phi(\boldsymbol{K}_i \boldsymbol{p}_t)) \boldsymbol{X}_i \boldsymbol{v} \boldsymbol{g}_i(\boldsymbol{p}_t)\|_2
\end{aligned}
$$

Next, we bound each term $A_i = \|\boldsymbol{J}(\phi(\boldsymbol{K}_i \boldsymbol{p}_{t+\tau})) \boldsymbol{X}_i \boldsymbol{v} \boldsymbol{g}_i(\boldsymbol{p}_{t+\tau}) - \boldsymbol{J}(\phi(\boldsymbol{K}_i \boldsymbol{p}_t)) \boldsymbol{X}_i \boldsymbol{v} \boldsymbol{g}_i(\boldsymbol{p}_t)\|_2$ in the sum as follows:

$$
\begin{aligned}
A_i &= \|(\boldsymbol{J}(\phi(\boldsymbol{K}_i \boldsymbol{p}_{t+\tau})) - \boldsymbol{J}(\phi(\boldsymbol{K}_i \boldsymbol{p}_t))) \boldsymbol{X}_i \boldsymbol{v} \boldsymbol{g}_i(\boldsymbol{p}_{t+\tau}) + \boldsymbol{J}(\phi(\boldsymbol{K}_i \boldsymbol{p}_t)) \boldsymbol{X}_i \boldsymbol{v}(\boldsymbol{g}_i(\boldsymbol{p}_{t+\tau}) - \boldsymbol{g}_i(\boldsymbol{p}_t)))\|_2 \\
&\leq \|(\boldsymbol{J}(\phi(\boldsymbol{K}_i \boldsymbol{p}_{t+\tau})) - \boldsymbol{J}(\phi(\boldsymbol{K}_i \boldsymbol{p}_t))) \boldsymbol{X}_i \boldsymbol{v} \boldsymbol{g}_i(\boldsymbol{p}_{t+\tau})\|_2 + \|\boldsymbol{J}(\phi(\boldsymbol{K}_i \boldsymbol{p}_t)) \boldsymbol{X}_i \boldsymbol{v}(\boldsymbol{g}_i(\boldsymbol{p}_{t+\tau}) - \boldsymbol{g}_i(\boldsymbol{p}_t)))\|_2 \\
&\leq C_\sigma \|\boldsymbol{v}\|_2 (\|\boldsymbol{J}(\phi(\boldsymbol{K}_i \boldsymbol{p}_{t+\tau})) - \boldsymbol{J}(\phi(\boldsymbol{K}_i \boldsymbol{p}_t))\|_2 \|\boldsymbol{g}_i(\boldsymbol{p}_t)\|_2 + \|\boldsymbol{J}(\phi(\boldsymbol{K}_i \boldsymbol{p}_{t+\tau}))\|_2 \|\boldsymbol{g}_i(\boldsymbol{p}_{t+\tau}) - \boldsymbol{g}_i(\boldsymbol{p}_t)\|_2)
\end{aligned}
$$

And we have:

$$
\|\boldsymbol{J}(\phi(\boldsymbol{K}_i \boldsymbol{p}_{t+\tau})) - \boldsymbol{J}(\phi(\boldsymbol{K}_i \boldsymbol{p}_t))\|_2 \leq 3C_\sigma \sigma_{max}(\boldsymbol{W}) \|\boldsymbol{p}_{t+\tau} - \boldsymbol{p}_t\|_2,
\tag{25}
$$

$$\|\boldsymbol{g}_i(\boldsymbol{p}_t)\|_2 \leq L\sqrt{\frac{2(\ell(f_{t+\tau}^i) - \ell^*)}{m}}, \quad \text{(Strong convexity and smoothness of } \ell\text{)} \tag{26}$$

$$\|J(\phi(\boldsymbol{K}_i\boldsymbol{p}_{+\tau}))\|_2 \leq 1, \quad \text{(Lemma 1)} \tag{27}$$

$$\|\boldsymbol{g}_i(\boldsymbol{p}_{t+\tau}) - \boldsymbol{g}_i(\boldsymbol{p}_t)\|_2 \leq L\|f_{t+\tau}^i - f_t^i\|_2, \quad \text{(Lipschitz continuity of the gradient of } \ell\text{)}$$
$$\leq LC_\sigma\|\boldsymbol{v}\|_2\|\phi(\boldsymbol{K}_i\boldsymbol{p}_{t+\tau}) - \phi(\boldsymbol{K}_i\boldsymbol{p}_t)\|_2$$
$$\leq 2LC_\sigma^2\sigma_{max}(\boldsymbol{W})\|\boldsymbol{v}\|_2\|\boldsymbol{p}_{t+\tau} - \boldsymbol{p}_t\|_2. \tag{28}$$

We can conclude that

$$\|\nabla_{\boldsymbol{p}}\mathcal{L}(\boldsymbol{p}_{t+\tau}) - \nabla_{\boldsymbol{p}}\mathcal{L}(\boldsymbol{p}_t)\|_2 \leq C_\sigma^2\sigma_{max}(\boldsymbol{W})\|\boldsymbol{v}\|_2\Big(2LC_\sigma^2\sigma_{max}(\boldsymbol{W})\|\boldsymbol{v}\|_2 \tag{29}$$

$$+ 3C_\sigma\sigma_{max}(\boldsymbol{W})L\sum_{i=1}^N\sqrt{\frac{2\ell(f_{t+\tau}^i) - \ell^*}{mN^2}}\Big)\|\boldsymbol{p}_{t+\tau} - \boldsymbol{p}_t\|_2$$

$$\leq C_\sigma^3\sigma^2(\boldsymbol{W})\|\boldsymbol{v}\|_2(2LC_\sigma\|\boldsymbol{v}\|_2 + 3L\sqrt{\frac{2(\mathcal{L}_0 - \mathcal{L}^*)}{m}})\|\boldsymbol{p}_{t+\tau} - \boldsymbol{p}_t\|_2$$

$$= L'\|\boldsymbol{p}_{t+\tau} - \boldsymbol{p}_t\|_2 \tag{30}$$

Finally, we determine a uniform lower bound for $\sigma_{min}(\mathbf{F}(\boldsymbol{p}_t))$ to establish the PL inequality.

We consider the deviation of $\boldsymbol{F}(\boldsymbol{p}_t)$ from its initialization:

$$\|\boldsymbol{F}(\boldsymbol{p}_t) - \boldsymbol{F}(\boldsymbol{p}_0)\|_F^2 = \frac{1}{N^2}\sum_{i=1}^N\|\boldsymbol{K}_i^\top\boldsymbol{J}(\phi(\boldsymbol{K}_i\boldsymbol{p}_t))\boldsymbol{X}_i\boldsymbol{v} - \boldsymbol{K}_i^\top\boldsymbol{J}(\phi(\boldsymbol{K}_i\boldsymbol{p}_0))\boldsymbol{X}_i\boldsymbol{v}\|_2^2$$

$$\leq \frac{C_\sigma^4\sigma_{max}^2(\boldsymbol{W})\|\boldsymbol{v}\|_2}{N^2}\sum_{i=1}^N\|\boldsymbol{J}(\phi(\boldsymbol{K}_i\boldsymbol{p}_t)) - \boldsymbol{J}(\phi(\boldsymbol{K}_i\boldsymbol{p}_0))\|_2^2$$

Each term in the sum has the following bound:

$$\|\boldsymbol{J}(\phi(\boldsymbol{K}_i\boldsymbol{p}_t)) - \boldsymbol{J}(\phi(\boldsymbol{K}_i\boldsymbol{p}_0))\|_2 \leq 3C_\sigma\sigma_{max}(\boldsymbol{W})\|\boldsymbol{p}_t - \boldsymbol{p}_0\|_2$$
$$\leq \min(2, 3C_\sigma\sigma_{max}(\boldsymbol{W})C_p)$$

Let $\gamma = \frac{1}{\sqrt{N}}(\min(2, 3C_\sigma\sigma_{max}(\boldsymbol{W})C_p)C_\sigma^2\sigma_{max}(\boldsymbol{W})\|\boldsymbol{v}\|_2)$, we have

$$\|\boldsymbol{F}(\boldsymbol{p}_t) - \boldsymbol{F}(\boldsymbol{p}_0)\|_F \leq \gamma.$$

Using Weyl's inequality, we conclude that

$$\sigma_{min}(\boldsymbol{F}(\boldsymbol{p}_t)) \geq \sigma_{min}(\boldsymbol{F}(\boldsymbol{p}_0)) - \gamma > \sqrt{\mu/2m}. \tag{31}$$

Therefore, since the step size satisfies $\eta < \frac{2}{L'}$, we have

$$\mathcal{L}_{t+1} - \mathcal{L}^* \leq \mathcal{L}_t - \frac{\eta}{2}\|\nabla_\theta\mathcal{L}_t\|_2^2 - \mathcal{L}^*$$
$$\leq (1 - \eta\mu)(\mathcal{L}_t - \mathcal{L}^*)$$
$$\leq (1 - \eta\mu)^{t+1}(\mathcal{L}_0 - \mathcal{L}^*). \tag{32}$$

which concludes our proof. $\qquad\square$

Next, we use invoke the following lemma which we prove below:

**Lemma 5.** *Suppose that the data samples $\boldsymbol{X}_i$ are drawn from a continuous distribution $\mathcal{D}$. Let $\boldsymbol{v} = \alpha\mathbf{u}$ with $\|\mathbf{u}\|_2 = 1$. There exists $A > 0$ such that if $\alpha > A$, then $\mathbb{P}_{\mathcal{D}}(\sigma_{min}(\mathbf{F}(\boldsymbol{p}_0)) > \gamma + \sqrt{\mu/2m})) = 1$.*

*Proof.* First, we analyze the bound $\gamma + \sqrt{\frac{\mu}{2m}}$ as a function of $\|\boldsymbol{v}\|_2 = \alpha$.

For very large $\alpha$, we have $L' = C_1\alpha^2 + C_2\alpha$, therefore $\mu = O(1/\alpha^2)$ and $C_p = O(1/\alpha^2)$. Moreover, when $C_p$ is sufficiently small, we have $3C_\sigma\sigma_{max}(\boldsymbol{W})C_p \leq 2$ and $\gamma = O(1/\alpha)$.

Next, we analyze $\sigma_{min}(\boldsymbol{F}(\boldsymbol{p}_0))$. We start by showing that $\sigma_{min}(\boldsymbol{F}(\boldsymbol{p}_0)) > 0$, then we show that $\sigma_{min}(\boldsymbol{F}(\boldsymbol{p}_0)) = O(\alpha)$ to conclude that when $\alpha$ is large enough, the condition is satisfied.

- First, we show that all rows of $\boldsymbol{F}(\boldsymbol{p}_0)$ are non zero. We have $\boldsymbol{W}$ has full rank and $\boldsymbol{X}_i$ has full row for all $i = 1, \ldots, N$. As a result, $X_i^T$ is injective for all $i$ and

$$\boldsymbol{J}(\phi(\boldsymbol{K}_i\boldsymbol{p}_0))\boldsymbol{X}_i\boldsymbol{v} \neq 0 \implies \boldsymbol{K}_i^\top\boldsymbol{J}(\phi(\boldsymbol{K}_i\boldsymbol{p}_0)\boldsymbol{X}_i\boldsymbol{v} \neq 0. \tag{33}$$

  Moreover, we know that $\ker \boldsymbol{J}(\phi(\boldsymbol{K}_i\boldsymbol{p}_0)) = \mathrm{span}(\mathbb{1}_n)$, therefore for all $i = 1, \ldots, N$, we need to ensure that $X_i\boldsymbol{v} \notin \ker \boldsymbol{J}(\phi(\boldsymbol{K}_i\boldsymbol{p}_0))$. However, if we assume that the $X_i$'s are drawn from a distribution $\mathcal{D}$, then the preimage of $\ker \boldsymbol{J}(\phi(\boldsymbol{K}_i\boldsymbol{p}_0))$ in $\mathbb{R}^d$ has measure 0. Therefore, any choice of vector $\boldsymbol{v}$ will satisfy the condition with probability 1.

- Next, we show that the rows of $\boldsymbol{F}(\boldsymbol{p}_0)$ are linearly independent with probability 1. Since $W$ has full rank and each $\boldsymbol{X}_i$ has full row rank, every row of $\boldsymbol{F}(\boldsymbol{p}_0)$ takes the form

$$r_i = \boldsymbol{K}_i^\top\boldsymbol{J}(\phi(\boldsymbol{K}_i\boldsymbol{p}_0))\boldsymbol{X}_i\boldsymbol{u}, \qquad \boldsymbol{u} = \boldsymbol{v}/\|\boldsymbol{v}\|,$$

  scaled by $\alpha$. Thus $r_i$ is a non-degenerate random vector in $\mathbb{R}^d$ whenever $\boldsymbol{X}_i$ is drawn from a continuous distribution. If the rows $\{r_1, \ldots, r_N\}$ were linearly dependent, there would exist a nonzero $\lambda \in \mathbb{R}^N$ such that $\sum_{i=1}^{N} \lambda_i r_i = 0$. This relation defines a nontrivial polynomial constraint on the entries of $\{\boldsymbol{X}_i\}$, which holds on an algebraic variety of Lebesgue measure zero. Therefore, with probability 1 under the data distribution, the rows of $\boldsymbol{F}(\boldsymbol{p}_0)$ are linearly independent whenever $d \geq N$, and hence $\sigma_{\min}(\boldsymbol{F}(\boldsymbol{p}_0)) > 0$.

- We can write

$$\boldsymbol{F}(\boldsymbol{p}_0) = \frac{\alpha}{N} \begin{bmatrix} (\boldsymbol{K}_1^\top\boldsymbol{J}(\phi(\boldsymbol{K}_1\boldsymbol{p}_0))\boldsymbol{X}_1\mathbf{u})^\top \\ \vdots \\ (\boldsymbol{K}_N^\top\boldsymbol{J}(\phi(\boldsymbol{K}_N\boldsymbol{p}_0))\boldsymbol{X}_N\mathbf{u})^\top \end{bmatrix} \tag{34}$$

  therefore $\sigma_{min}(\boldsymbol{F}(\boldsymbol{p}_0)) = \Theta(\alpha)$, which allows us to conclude that for $\alpha$ large enough, we have $\sigma_{min}(\boldsymbol{F}(\boldsymbol{p}_0)) > \gamma + \sqrt{\frac{\mu}{2m}}$ with probability 1.

$\square$

# F   PROOF OF LEMMA 3

*Proof.* Fix $i \in [N]$ and write the scores $s_{ij} := \langle \boldsymbol{K}_i[j,:], \boldsymbol{p}^\star \rangle$. The softmax attention is

$$(\boldsymbol{a}_i^\star)_j \;=\; \frac{e^{s_{ij}}}{\sum_{k=1}^n e^{s_{ik}}} \;=\; \frac{e^{s_{ij}}}{\sum_{k \in S_i} e^{s_{ik}} + \sum_{\ell \notin S_i} e^{s_{i\ell}}}.$$

Let $m_i := \min_{j \in S_i} s_{ij}$ and $M_i := \max_{j \in S_i} s_{ij}$, so that $\Gamma_i = M_i - m_i$. By definition of $\Delta_i$ we also have $\max_{\ell \notin S_i} s_{i\ell} \leq m_i - \Delta_i$.

**Lower bound for important tokens.**   For any $j \in S_i$,

$$(\boldsymbol{a}_i^\star)_j \;\geq\; \frac{e^{m_i}}{\sum_{k \in S_i} e^{s_{ik}} + \sum_{\ell \notin S_i} e^{s_{i\ell}}} \;\geq\; \frac{e^{m_i}}{e^{m_i} + (s_i - 1)e^{M_i} + (n - s_i)e^{m_i - \Delta_i}}.$$

Dividing numerator and denominator by $e^{m_i}$ yields

$$(\boldsymbol{a}_i^\star)_j \;\geq\; \frac{1}{1 + (s_i - 1)e^{\Gamma_i} + (n - s_i)e^{-\Delta_i}}.$$

Taking the minimum over $j \in S_i$ and then the minimum over $i$ gives

$$a_I \;\geq\; \frac{1}{1 + (s_{\max} - 1)e^{\Gamma_{\max}} + (n - s_{\min})e^{-\Delta_{\min}}},$$

which is equation 13.

**Upper bound for unimportant tokens.**   Fix $\ell \notin S_i$ and denote $u_i := s_{i\ell}$. By the definition of $\Delta_i$ we have $m_i \geq u_i + \Delta_i$, hence for any $j \in S_i$, $e^{s_{ij}} \geq e^{m_i} \geq e^{u_i + \Delta_i}$. Therefore

$$(\boldsymbol{a}_i^\star)_\ell = \frac{e^{u_i}}{e^{u_i} + \sum_{j \in S_i} e^{s_{ij}} + \sum_{\ell' \notin S_i, \, \ell' \neq \ell} e^{s_{i\ell'}}} \;\leq\; \frac{e^{u_i}}{e^{u_i} + s_i e^{u_i + \Delta_i}} = \frac{1}{1 + s_i e^{\Delta_i}}.$$

Taking the maximum over $\ell \notin S_i$ and then the maximum over $i$ yields

$$a_U \;\leq\; \frac{1}{1 + s_{\min} e^{\Delta_{\min}}},$$

which is equation 14.

**Sharper $a_U$ under uniform important scores.**   Assume now $\Gamma_i = 0$ for all $i$, i.e., $s_{ij} = m_i$ for every $j \in S_i$. For any $\ell \notin S_i$ we still have $u_i \leq m_i - \Delta_i$, hence

$$(\boldsymbol{a}_i^\star)_\ell \;=\; \frac{e^{u_i}}{s_i e^{m_i} + e^{u_i} + \sum_{\ell' \notin S_i, \, \ell' \neq \ell} e^{s_{i\ell'}}} \;\leq\; \frac{e^{m_i - \Delta_i}}{s_i e^{m_i} + (n - s_i)e^{m_i - \Delta_i}} \;=\; \frac{e^{-\Delta_i}}{s_i + (n - s_i)e^{-\Delta_i}}.$$

Taking the maximum over $\ell \notin S_i$ and then over $i$, and using $s_i \geq s_{\min}$ and $n - s_i \geq n - s_{\max}$ (so the denominator is bounded below by $s_{\min} + (n - s_{\max})e^{-\Delta_{\min}}$), we obtain

$$a_U \;\leq\; \frac{e^{-\Delta_{\min}}}{s_{\min} + (n - s_{\max})e^{-\Delta_{\min}}}.$$

This concludes the proof.   $\square$

# G    PROOF OF THEOREM 2

## G.1    DERIVATIVES AND BLOCK STRUCTURE OF $\mathcal{J}(\boldsymbol{a}^\star)$

Recall the gradient descent update equation:

$$\boldsymbol{p}_{t+1} = \boldsymbol{p}_t - \frac{\eta}{N} \sum_{j=1}^{N} \boldsymbol{K}_j^\top \boldsymbol{J}(\phi(\boldsymbol{K}_j \boldsymbol{p}_t)) \nabla \tilde{\ell}_j(\phi(\boldsymbol{K}_j \boldsymbol{p}_t)). \tag{35}$$

The induced dynamics on the attention vectors for each $i = 1, \ldots, N$ can be written as:

$$\phi(\boldsymbol{K}_i \boldsymbol{p}_{t+1}) = \phi\left(\boldsymbol{K}_i \boldsymbol{p}_t - \frac{\eta}{N} \sum_{j=1}^{N} \boldsymbol{K}_i \boldsymbol{K}_j^T \boldsymbol{J}(\phi(\boldsymbol{K}_j \boldsymbol{p}_t)) \nabla \tilde{\ell}_j(\phi(\boldsymbol{K}_j \boldsymbol{p}_t))\right) \tag{36}$$

We set $\boldsymbol{a}_i^t = \phi(\boldsymbol{K}_i \boldsymbol{p}_t)$. Next using Taylor's theorem, we can write

$$\boldsymbol{a}_i^{t+1} = \boldsymbol{a}_i^t - \frac{\eta}{N} \boldsymbol{J}(\boldsymbol{a}_i^t) \sum_{j=1}^{N} \boldsymbol{K}_i \boldsymbol{K}_j^T \boldsymbol{J}(\boldsymbol{a}_j^t) \nabla \tilde{\ell}_j(\boldsymbol{a}_j^t) + \mathcal{O}(\eta^2), \tag{37}$$

Let $V_i(\boldsymbol{a}) = \boldsymbol{J}(\boldsymbol{a}_i) \sum_{j=1}^{N} \boldsymbol{K}_i \boldsymbol{K}_j^T \boldsymbol{J}(\boldsymbol{a}_j) \nabla \tilde{\ell}_i(\boldsymbol{a}_j)$, where $\boldsymbol{a} = (\boldsymbol{a}_1, \ldots, \boldsymbol{a}_N)$. To linearize the system around $\boldsymbol{a}^*$, we write the dynamics of $\xi_t = \boldsymbol{a}_t - \boldsymbol{a}^*$:

$$\xi_{t+1} = \xi_t - \frac{\eta}{N} \boldsymbol{V}(\boldsymbol{a}^* + \xi_t) + \mathcal{O}(\eta^2) \tag{38}$$

The linearization of $\boldsymbol{V}(\boldsymbol{a}^* + \zeta_t)$ can be expressed as:

$$\boldsymbol{V}(\boldsymbol{a}^* + \zeta_t) = \boldsymbol{V}(\boldsymbol{a}^*) + \mathcal{J}(\boldsymbol{a}^*)\zeta_t = \mathcal{J}(\boldsymbol{a}^*)\zeta_t \tag{39}$$

Where we have used the fact that $\boldsymbol{V}(\boldsymbol{a}^*) = 0$ because $\boldsymbol{a}^*$ is a stationary point. Next, we will compute the derivatives of each $\boldsymbol{V}_i$ with respect to $\boldsymbol{a}_1, \ldots, \boldsymbol{a}_N$ using the product rule and the identity

$$\frac{d}{d\boldsymbol{x}}(\boldsymbol{J}(\boldsymbol{x})f(\boldsymbol{x})) = \boldsymbol{J}(\boldsymbol{x}) \, Df(\boldsymbol{x}) \, - \, \boldsymbol{x} f(\boldsymbol{x})^\top + \mathrm{diag}(f(\boldsymbol{x})) \, - \, (\boldsymbol{x}^\top f(\boldsymbol{x})) \, \boldsymbol{I}, \tag{40}$$

where the identity equation 40 follows from a direct application of the product rule. Indeed, since $\boldsymbol{J}(\boldsymbol{x}) = \mathrm{diag}(\boldsymbol{x}) - \boldsymbol{x}\boldsymbol{x}^\top$, we can write

$$\boldsymbol{J}(\boldsymbol{x})f(\boldsymbol{x}) = \mathrm{diag}(\boldsymbol{x})f(\boldsymbol{x}) - \boldsymbol{x}(\boldsymbol{x}^\top f(\boldsymbol{x})).$$

For the first term, using the componentwise product rule on $[\mathrm{diag}(\boldsymbol{x})f(\boldsymbol{x})]_i = x_i f_i(\boldsymbol{x})$ gives

$$\frac{d}{d\boldsymbol{x}}(\mathrm{diag}(\boldsymbol{x})f(\boldsymbol{x})) = \mathrm{diag}(\boldsymbol{x}) \, Df(\boldsymbol{x}) + \mathrm{diag}(f(\boldsymbol{x})).$$

For the second term, let $s(\boldsymbol{x}) = \boldsymbol{x}^\top f(\boldsymbol{x})$ so that the term is $\boldsymbol{x}s(\boldsymbol{x})$. Then $\nabla s(\boldsymbol{x}) = f(\boldsymbol{x}) + Df(\boldsymbol{x})^\top \boldsymbol{x}$, and another product rule yields

$$\frac{d}{d\boldsymbol{x}}(\boldsymbol{x}s(\boldsymbol{x})) = s(\boldsymbol{x}) \, \boldsymbol{I} + \boldsymbol{x} \nabla s(\boldsymbol{x})^\top = (\boldsymbol{x}^\top f(\boldsymbol{x})) \, \boldsymbol{I} + \boldsymbol{x} f(\boldsymbol{x})^\top + \boldsymbol{x}\boldsymbol{x}^\top Df(\boldsymbol{x}).$$

Subtracting these two derivatives and regrouping terms gives exactly equation 40.

If $i \neq k$, we have:

$$\frac{\partial \boldsymbol{V}_i(\boldsymbol{a})}{\partial \boldsymbol{a}_k} = \boldsymbol{J}(\boldsymbol{a}_i) \boldsymbol{K}_i \boldsymbol{K}_k^T \frac{d}{d\boldsymbol{a}_k}(\boldsymbol{J}(\boldsymbol{a}_k) \nabla \tilde{\ell}_k(\boldsymbol{a}_k)) \tag{41}$$

Therefore,

$$\frac{\partial \mathbf{V}_i(\mathbf{a})}{\partial \mathbf{a}_k} = \mathbf{J}(\mathbf{a}_i)\mathbf{K}_i\mathbf{K}_k^T\left(\mathbf{J}(\mathbf{a}_k)\nabla^2\tilde{\ell}_k(\mathbf{a}_k) - \mathbf{a}_k\nabla\tilde{\ell}_k(\mathbf{a}_k)^T + \mathrm{diag}(\nabla\tilde{\ell}_k(\mathbf{a}_k)) - \mathbf{a}_k^T\nabla\tilde{\ell}_k(\mathbf{a}_k)I\right) \tag{42}$$

For $i = k$, we have:

$$\frac{\partial \mathbf{V}_i(\mathbf{a})}{\partial \mathbf{a}_i} = \mathbf{J}(\mathbf{a}_i)\frac{\partial}{\partial \mathbf{a}_i}\left(\sum_{j=1}^{N}\mathbf{K}_i\mathbf{K}_j^T\mathbf{J}(\mathbf{a}_j)\nabla\tilde{\ell}_j(\mathbf{a}_j)\right)$$

$$- \mathbf{a}_i\left(\sum_{j=1}^{N}\mathbf{K}_i\mathbf{K}_j^T\mathbf{J}(\mathbf{a}_j)\nabla\tilde{\ell}_j(\mathbf{a}_j)\right)^T + \mathrm{diag}\left(\sum_{j=1}^{N}\mathbf{K}_i\mathbf{K}_j^T\mathbf{J}(\mathbf{a}_j)\nabla\tilde{\ell}_j(\mathbf{a}_j)\right) - \mathbf{a}_i^T\left(\sum_{j=1}^{N}\mathbf{K}_i\mathbf{K}_j^T\mathbf{J}(\mathbf{a}_j)\nabla\tilde{\ell}_j(\mathbf{a}_j)\right)I$$

$$= \mathbf{J}(\mathbf{a}_i)\mathbf{K}_i\mathbf{K}_i^T\left(\mathbf{J}(\mathbf{a}_i)\nabla^2\tilde{\ell}_i(\mathbf{a}_i) - \mathbf{a}_i\nabla\tilde{\ell}_i(\mathbf{a}_i)^T + \mathrm{diag}(\nabla\tilde{\ell}_i(\mathbf{a}_i)) - \mathbf{a}_i^T\nabla\tilde{\ell}_i(\mathbf{a}_i)I\right)$$

$$- \mathbf{a}_i\left(\sum_{j=1}^{N}\mathbf{K}_i\mathbf{K}_j^T\mathbf{J}(\mathbf{a}_j)\nabla\tilde{\ell}_j(\mathbf{a}_j)\right)^T + \mathrm{diag}\left(\sum_{j=1}^{N}\mathbf{K}_i\mathbf{K}_j^T\mathbf{J}(\mathbf{a}_j)\nabla\tilde{\ell}_j(\mathbf{a}_j)\right) - \mathbf{a}_i^T\left(\sum_{j=1}^{N}\mathbf{K}_i\mathbf{K}_j^T\mathbf{J}(\mathbf{a}_j)\nabla\tilde{\ell}_j(\mathbf{a}_j)\right)I$$

Since $\mathbf{a}^*$ is a stationary point, we have $\sum_{j=1}^{N}\mathbf{K}_j^T\mathbf{J}(\mathbf{a}_j^*)\nabla\tilde{\ell}_j(\mathbf{a}_j^*) = 0$ which reduces $\frac{\partial \mathbf{V}_i(\mathbf{a}^*)}{\partial \mathbf{a}_i}$ to:

$$\frac{\partial \mathbf{V}_i(\mathbf{a}^*)}{\partial \mathbf{a}_i} = \mathbf{J}(\mathbf{a}_i^*)\mathbf{K}_i\mathbf{K}_i^T\left(\mathbf{J}(\mathbf{a}_i^*)\nabla^2\tilde{\ell}_i(\mathbf{a}_i^*) - \mathbf{a}_i^*\nabla\tilde{\ell}_i(\mathbf{a}_i^*)^T + \mathrm{diag}(\nabla\tilde{\ell}_i(\mathbf{a}_i^*)) - \mathbf{a}_i^{*T}\nabla\tilde{\ell}_i(\mathbf{a}_i^*)I\right)$$

Therefore, we can write for all $i, k \in \{1, \ldots, N\}$

$$\frac{\partial \mathbf{V}_i(\mathbf{a}^*)}{\partial \mathbf{a}_k} = \mathbf{J}(\mathbf{a}_i^*)\mathbf{K}_i\mathbf{K}_k^T\left(\mathbf{J}(\mathbf{a}_k^*)\nabla^2\tilde{\ell}_k(\mathbf{a}_k^*) - \mathbf{a}_k^*\nabla\tilde{\ell}_k(\mathbf{a}_k^*)^T + \mathrm{diag}(\nabla\tilde{\ell}_k(\mathbf{a}_k^*)) - \mathbf{a}_k^{*T}\nabla\tilde{\ell}_k(\mathbf{a}_k^*)I\right)$$

Our analysis focuses on stationary points that globally minimize $\ell$, and hence also $\tilde{\ell}$, so that $\nabla\tilde{\ell}(\mathbf{a}^\star) = 0$ and the last three terms in the previous derivative expression vanish.

Collecting blocks gives the Jacobian

$$\mathcal{J}(\mathbf{a}^\star) = \begin{pmatrix} A_{11} & \cdots & A_{1N} \\ \vdots & \ddots & \vdots \\ A_{N1} & \cdots & A_{NN} \end{pmatrix}, \qquad A_{ik} = \mathbf{J}(\mathbf{a}_i^\star)\mathbf{K}_i\mathbf{K}_k^\top\mathbf{J}(\mathbf{a}_k^\star)\mathbf{H}_k, \tag{43}$$

with $\mathbf{H}_k := \nabla^2\tilde{\ell}_k(\mathbf{a}_k^\star) \succeq 0$.

## G.2 FACTORIZATION AND SPECTRAL EQUIVALENCE

Define

$$\mathbf{B} = \begin{bmatrix} \mathbf{J}(\mathbf{a}_1^\star)\mathbf{K}_1 \\ \vdots \\ \mathbf{J}(\mathbf{a}_N^\star)\mathbf{K}_N \end{bmatrix}, \qquad \mathbf{C} = \begin{bmatrix} \mathbf{K}_1^\top\mathbf{J}(\mathbf{a}_1^\star)\mathbf{H}_1 & \cdots & \mathbf{K}_N^\top\mathbf{J}(\mathbf{a}_N^\star)\mathbf{H}_N \end{bmatrix}.$$

Then $\mathcal{J}(\mathbf{a}^\star) = \mathbf{BC}$ and

$$\mathbf{M} := \mathbf{CB} = \sum_{i=1}^{N}\mathbf{K}_i^\top\mathbf{J}(\mathbf{a}_i^\star)\mathbf{H}_i\mathbf{J}(\mathbf{a}_i^\star)\mathbf{K}_i \succeq 0. \tag{44}$$

Since $\mathbf{BC}$ and $\mathbf{CB}$ share the same multiset of nonzero eigenvalues, the nonzero spectrum of $\mathcal{J}(\mathbf{a}^\star)$ is real and equal to that of $\mathbf{M}$.

### G.3 BOUNDS UNDER SPARSE ATTENTION AND ALIGNMENT

First, note that the conditioning of $\mathbf{M}$ depends not only on the alignment of the keys ($\boldsymbol{K}_i$) but also on the alignment of the value vectors $\mathbf{u}_i := \boldsymbol{X}_i \boldsymbol{v}$ with the important tokens.

We have $\boldsymbol{H}_i = \ell''(s_i^\star, y_i)\,\mathbf{u}_i \mathbf{u}_i^\top$, which satisfies $m\,\mathbf{u}_i \mathbf{u}_i^\top \preceq \boldsymbol{H}_i \preceq L\,\mathbf{u}_i \mathbf{u}_i^\top$. The quadratic form for $\mathbf{M}$ is therefore

$$\mathbf{x}^\top \mathbf{M} \mathbf{x} = \sum_{i=1}^{N} \ell''(s_i^\star, y_i) \|\mathbf{u}_i^\top \boldsymbol{J}(\boldsymbol{a}_i^\star) \boldsymbol{K}_i \mathbf{x}\|^2. \tag{45}$$

This leads to the matrix sandwich inequality:

$$m\,\widetilde{\mathbf{M}} \preceq \mathbf{M} \preceq L\,\widetilde{\mathbf{M}}, \quad \text{where} \quad \widetilde{\mathbf{M}} := \sum_{i=1}^{N} \big(\boldsymbol{K}_i^\top \boldsymbol{J}(\boldsymbol{a}_i^\star)\mathbf{u}_i\big)\big(\mathbf{u}_i^\top \boldsymbol{J}(\boldsymbol{a}_i^\star)\boldsymbol{K}_i\big). \tag{46}$$

We will now bound the extremal eigenvalues of $\widetilde{\mathbf{M}}$ by choosing test vectors $\mathbf{x}$ from the subspaces $I$ and $U$.

**Lemma 6** (Lower bound on $\lambda_{\max}(\widetilde{\boldsymbol{M}})$ under value–compatibility and invertible $\boldsymbol{W}$). *Let $\widetilde{\boldsymbol{M}} = \sum_{i=1}^{N} \big(\boldsymbol{K}_i^\top \boldsymbol{J}(\boldsymbol{a}_i^\star)\boldsymbol{u}_i\big)\big(\boldsymbol{u}_i^\top \boldsymbol{J}(\boldsymbol{a}_i^\star)\boldsymbol{K}_i\big)$ with $\boldsymbol{u}_i = \boldsymbol{X}_i \boldsymbol{v}$. Under assumption B, we have*

$$\lambda_{\max}(\widetilde{\boldsymbol{M}}) \geq \frac{a_I^2}{\|\boldsymbol{W}^{-1}\boldsymbol{v}\|_2^2} \sum_{i=1}^{N} s_i^2\,\delta_i^4.$$

*In particular, if $s_i \geq s_{\min}$ and $\delta_i \geq \delta_{\min} > 0$ for all $i$,*

$$\lambda_{\max}(\widetilde{\boldsymbol{M}}) \geq \frac{N\,a_I^2\,s_{\min}^2\,\delta_{\min}^4}{\|\boldsymbol{W}^{-1}\boldsymbol{v}\|_2^2} \geq \frac{N\,a_I^2\,s_{\min}^2\,\delta_{\min}^4\,\sigma_{\min}^2(\boldsymbol{W})}{\|\boldsymbol{v}\|_2^2},$$

*where $\sigma_{\min}(\boldsymbol{W})$ denotes the smallest singular value of $\boldsymbol{W}$.*

*Proof. Choice of the test vector from invertibility.* Since $\boldsymbol{W}$ is invertible and $\boldsymbol{K}_i = \boldsymbol{X}_i \boldsymbol{W}$ while $\boldsymbol{u}_i = \boldsymbol{X}_i \boldsymbol{v}$, the unique vector

$$\boldsymbol{x} := \boldsymbol{W}^{-1}\boldsymbol{v}$$

satisfies the alignment identity $\boldsymbol{K}_i \boldsymbol{x} = \boldsymbol{X}_i \boldsymbol{W}\boldsymbol{W}^{-1}\boldsymbol{v} = \boldsymbol{X}_i \boldsymbol{v} = \boldsymbol{u}_i$ for all $i$. Let $\boldsymbol{x} = \boldsymbol{x}/\|\boldsymbol{x}\|_2$.

By definition of $\widetilde{\boldsymbol{M}}$ and the alignment $\boldsymbol{K}_i \boldsymbol{x} = \boldsymbol{u}_i/\|\boldsymbol{x}\|_2$,

$$\boldsymbol{x}^\top \widetilde{\boldsymbol{M}} \boldsymbol{x} = \sum_{i=1}^{N} \big(\boldsymbol{u}_i^\top \boldsymbol{J}(\boldsymbol{a}_i^\star)\boldsymbol{K}_i \boldsymbol{x}\big)^2 = \frac{1}{\|\boldsymbol{x}\|_2^2} \sum_{i=1}^{N} \big(\boldsymbol{u}_i^\top \boldsymbol{J}(\boldsymbol{a}_i^\star)\boldsymbol{u}_i\big)^2.$$

Using the variance identity $\boldsymbol{u}^\top \boldsymbol{J}(\boldsymbol{z})\boldsymbol{u} = \sum_j \boldsymbol{a}_j\big((\boldsymbol{u})_j - \boldsymbol{a}^\top \boldsymbol{u}\big)^2$ with $\mu_i = \boldsymbol{a}_i^{\star\top}\boldsymbol{u}_i$,

$$\boldsymbol{u}_i^\top \boldsymbol{J}(\boldsymbol{a}_i^\star)\boldsymbol{u}_i = \sum_{j=1}^{n} (\boldsymbol{a}_i^\star)_j\big((\boldsymbol{u}_i)_j - \mu_i\big)^2 \geq \sum_{j \in S_i} (\boldsymbol{a}_i^\star)_j\,\delta_i^2 \geq s_i\,a_I\,\delta_i^2.$$

Squaring and summing over $i$ yields

$$\sum_{i=1}^{N} \big(\boldsymbol{u}_i^\top \boldsymbol{J}(\boldsymbol{a}_i^\star)\boldsymbol{u}_i\big)^2 \geq \sum_{i=1}^{N} a_I^2\,s_i^2\,\delta_i^4.$$

Therefore,

$$\boldsymbol{x}^\top \widetilde{\boldsymbol{M}} \boldsymbol{x} \geq \frac{a_I^2}{\|\boldsymbol{x}\|_2^2} \sum_{i=1}^{N} s_i^2\,\delta_i^4 = \frac{a_I^2}{\|\boldsymbol{W}^{-1}\boldsymbol{v}\|_2^2} \sum_{i=1}^{N} s_i^2\,\delta_i^4.$$

By the Courant–Fischer theorem, $\lambda_{\max}(\widetilde{\boldsymbol{M}}) \geq \boldsymbol{x}^\top \widetilde{\boldsymbol{M}}\,\boldsymbol{x}$, proving the stated bound.

For the singular-value form, use $\|\boldsymbol{W}^{-1}\boldsymbol{v}\|_2 \leq \|\boldsymbol{W}^{-1}\|\,\|\boldsymbol{v}\|_2 = \|\boldsymbol{v}\|_2/\sigma_{\min}(\boldsymbol{W})$, which gives

$$\frac{1}{\|\boldsymbol{W}^{-1}\boldsymbol{v}\|_2^2} \geq \frac{\sigma_{\min}^2(\boldsymbol{W})}{\|\boldsymbol{v}\|_2^2}.$$

Substituting this into the previous display yields the looser but convenient expression in terms of $\sigma_{\min}(\boldsymbol{W})$ and $\|\boldsymbol{v}\|_2$. $\qquad\square$

**Remark 8** (Why $\boldsymbol{x} = \boldsymbol{W}^{-1}\boldsymbol{v}$ is canonical). *The choice $\boldsymbol{x} = \boldsymbol{W}^{-1}\boldsymbol{v}$ is forced by invertibility: it is the* unique *vector whose key projection matches the value signal on every example, $\boldsymbol{K}_i\boldsymbol{x} = \boldsymbol{u}_i$. This collapses each quadratic term to the z-variance of $\boldsymbol{u}_i$ and eliminates cross-couplings due to $\boldsymbol{K}_i$, making the bound depend only on the head mass $a_I$, the head sizes $s_i$, the value–compatibility margins $\delta_i$, and the conditioning of $\boldsymbol{W}$ via $\|\boldsymbol{W}^{-1}\boldsymbol{v}\|_2$ (or $\sigma_{\min}(\boldsymbol{W})$).*

**Lemma 7** (Upper bound on $\lambda_{\min}(\widetilde{\mathbf{M}})$ under per-example leakage). *Under assumptions A, B, we have*
$$\lambda_{\min}(\widetilde{\mathbf{M}}) \;\leq\; C_v\, A_\star\, \lambda_{\max}(\boldsymbol{P}_U \Sigma_U \boldsymbol{P}_U).$$

*Proof.* Fix a unit $x_U \in U$ and set $w_i = \boldsymbol{K}_i x_U$. Split
$$w_{i,T} := (w_i)_{S_i^c} = \boldsymbol{K}_{i,S_i^c}^U x_U, \qquad w_{i,S} := (w_i)_{S_i} = \boldsymbol{K}_{i,S_i}^U x_U,$$
so that by the leakage dominance assumption $\|w_{i,S}\|_2 \leq \rho\,\|w_{i,T}\|_2$ for all $i$. By definition of $\widetilde{\mathbf{M}}$ and Cauchy–Schwarz,
$$x_U^\top \widetilde{\mathbf{M}} x_U = \sum_{i=1}^N \left(\mathbf{u}_i^\top \boldsymbol{J}(\boldsymbol{a}_i^\star) w_i\right)^2 \;\leq\; \sum_{i=1}^N \|\mathbf{u}_i\|_2^2\, \|\boldsymbol{J}(\boldsymbol{a}_i^\star) w_i\|_2^2 \;\leq\; C_v \sum_{i=1}^N \|\boldsymbol{J}(\boldsymbol{a}_i^\star) w_i\|_2^2.$$
Using $\boldsymbol{J}(z) = \operatorname{diag}(z) - zz^\top$ and expanding,
$$\|\boldsymbol{J}(\boldsymbol{a}_i^\star) w_i\|_2^2 = \left\|\operatorname{diag}(\boldsymbol{a}_i^\star) w_i\right\|_2^2 + \left\|\boldsymbol{a}_i^\star \boldsymbol{a}_i^{\star\top} w_i\right\|_2^2 - w_i^\top\!\left(\operatorname{diag}(\boldsymbol{a}_i^\star)\boldsymbol{a}_i^\star \boldsymbol{a}_i^{\star\top} + \boldsymbol{a}_i^\star \boldsymbol{a}_i^{\star\top} \operatorname{diag}(\boldsymbol{a}_i^\star)\right) w_i$$
$$\leq T_{1,i} + T_{2,i} + T_{3,i},$$
where we bound each term by a multiple of $\|w_{i,T}\|_2^2$.

**1) Diagonal term.** On $S_i^c$ we have $\|(\boldsymbol{a}_i^\star)_{S_i^c}\|_\infty \leq a_U$, and $(\boldsymbol{a}_i^\star)_j \leq 1$ on $S_i$. Hence
$$T_{1,i} = \|\operatorname{diag}(\boldsymbol{a}_i^\star) w_i\|_2^2 = \sum_{j \in S_i} (\boldsymbol{a}_{ij}^\star)^2 w_{ij}^2 + \sum_{j \in S_i^c} (\boldsymbol{a}_{ij}^\star)^2 w_{ij}^2 \leq \|w_{i,S}\|_2^2 + a_U^2 \|w_{i,T}\|_2^2 \leq (\rho^2 + a_U^2)\,\|w_{i,T}\|_2^2.$$

**2) Rank-one term.** We have $\|\boldsymbol{a}_i^\star \boldsymbol{a}_i^{\star\top} w_i\|_2^2 = (\boldsymbol{a}_i^{\star\top} w_i)^2 \|\boldsymbol{a}_i^\star\|_2^2 \leq (\boldsymbol{a}_i^{\star\top} w_i)^2$. Split
$$\boldsymbol{a}_i^{\star\top} w_i = (\boldsymbol{a}_{i,S}^\star)^\top w_{i,S} + (\boldsymbol{a}_{i,T}^\star)^\top w_{i,T}.$$
Then
$$|(\boldsymbol{a}_{i,S}^\star)^\top w_{i,S}| \leq \|w_{i,S}\|_2 \leq \rho\,\|w_{i,T}\|_2, \qquad |(\boldsymbol{a}_{i,T}^\star)^\top w_{i,T}| \leq \|(\boldsymbol{a}_{i,T}^\star)\|_2 \|w_{i,T}\|_2 \leq \sqrt{m_i}\, a_U\, \|w_{i,T}\|_2,$$
where $m_i = n - s_i$ and we used $\|v\|_2 \leq \sqrt{m}\|v\|_\infty$ on $S_i^c$. Thus
$$T_{2,i} \leq \left(\rho_U + \sqrt{m_i}\, a_U\right)^2 \|w_{i,T}\|_2^2 \;\leq\; \left(2\rho^2 + 2m_i a_U^2\right) \|w_{i,T}\|_2^2.$$

**3) Cross term.** Using $w^\top (\operatorname{diag}(z)\, zz^\top + zz^\top \operatorname{diag}(z)) w = 2\,(z^\top w)\,((z \odot z)^\top w)$ and the same splits,
$$|z^\top w| \leq \rho\,\|w_{i,T}\|_2 + \sqrt{m_i}\, a_U\, \|w_{i,T}\|_2, \quad |(z \odot z)^\top w| \leq \rho\,\|w_{i,T}\|_2 + \sqrt{m_i}\, a_U^2\, \|w_{i,T}\|_2,$$
since $\|z_S \odot z_S\|_2 \leq \|z_S\|_2 \leq 1$ and $\|(z_T \odot z_T)\|_\infty \leq a_U^2$. Therefore,
$$T_{3,i} \leq 2\left(\rho_U + \sqrt{m_i}\, a_U\right)\left(\rho_U + \sqrt{m_i}\, a_U^2\right) \|w_{i,T}\|_2^2 \;\leq\; 4\left(\rho^2 + m_i a_U^2\right) \|w_{i,T}\|_2^2,$$
using $a_U^2 \leq a_U$ and $(a + b)^2 \leq 2(a^2 + b^2)$.

**Collecting.** Summing the three bounds and using $m_i \leq m_\star := n - s_{\min}$,
$$\|\boldsymbol{J}(\boldsymbol{a}_i^\star) w_i\|_2^2 \leq \left[(\rho^2 + a_U^2) + (2\rho^2 + 2m_i a_U^2) + (4\rho^2 + 4m_i a_U^2)\right] \|w_{i,T}\|_2^2 \leq 7\left(\rho^2 + m_\star a_U^2\right) \|w_{i,T}\|_2^2.$$
Therefore,
$$x_U^\top \widetilde{\mathbf{M}} x_U \leq C_v \sum_{i=1}^N 7\left(\rho^2 + m_\star a_U^2\right) \|w_{i,T}\|_2^2 = C_v\, A_\star\, x_U^\top \Sigma_U x_U,$$
with $A_\star = 7(\rho^2 + m_\star a_U^2)$. Taking $x_U$ to be a top eigenvector of $\boldsymbol{P}_U \Sigma_U \boldsymbol{P}_U$ yields the eigenvalue bound. $\qquad\square$

# H    PROOF OF COROLLARY 1

*Proof.* We consider the local neighborhood where the assumptions of Theorem 2 ensure that the gradient linearizes with a fixed symmetric positive semidefinite operator $\boldsymbol{H}_p$ at $\boldsymbol{p}^*$, so that the gradient descent (GD) error dynamics are

$$\boldsymbol{e}_{t+1} \;=\; \boldsymbol{e}_t - \eta\,\boldsymbol{H}_p\,\boldsymbol{e}_t \;=\; (\boldsymbol{I} - \eta\,\boldsymbol{H}_p)\,\boldsymbol{e}_t, \qquad \boldsymbol{e}_t := \boldsymbol{p}_t - \boldsymbol{p}^*. \tag{47}$$

Let $\{(\lambda_k, \boldsymbol{v}_k)\}_{k=1}^d$ be an eigen-decomposition of $\boldsymbol{H}_p$ with $0 \le \lambda_{\min} = \lambda_1 \le \cdots \le \lambda_{\max} = \lambda_d$ and orthonormal eigenvectors $\{\boldsymbol{v}_k\}$. Expand $\boldsymbol{e}_0$ in this basis:

$$\boldsymbol{e}_0 \;=\; \sum_{k=1}^d \alpha_k \boldsymbol{v}_k, \qquad \alpha_k := \boldsymbol{v}_k^\top \boldsymbol{e}_0.$$

Iterating equation 47 gives

$$\boldsymbol{e}_t \;=\; (\boldsymbol{I} - \eta\boldsymbol{H}_p)^t \boldsymbol{e}_0 \;=\; \sum_{k=1}^d (1 - \eta\lambda_k)^t \alpha_k \boldsymbol{v}_k.$$

Projecting onto the slowest mode $\boldsymbol{v}_{\min} := \boldsymbol{v}_1$ yields the exact evolution of that component:

$$\boldsymbol{e}_t^\top \boldsymbol{v}_{\min} \;=\; (1 - \eta\lambda_{\min})^t \, \alpha_1 \;=\; (1 - \eta\lambda_{\min})^t \, (\boldsymbol{e}_0^\top \boldsymbol{v}_{\min}),$$

which proves the claimed identity.

For the worst-case lower bound on the norm, apply $\|\boldsymbol{e}_t\|_2^2 \ge (\boldsymbol{e}_t^\top \boldsymbol{v}_{\min})^2$ to obtain

$$\|\boldsymbol{e}_t\|_2^2 \;\ge\; (1 - \eta\lambda_{\min})^{2t} \, (\boldsymbol{e}_0^\top \boldsymbol{v}_{\min})^2 \;=\; (1 - \eta\lambda_{\min})^{2t} \, \|\boldsymbol{e}_0\|_2^2 \cos^2(\theta_0),$$

where $\cos^2(\theta_0) := (\boldsymbol{e}_0^\top \boldsymbol{v}_{\min})^2 / \|\boldsymbol{e}_0\|_2^2$ is the initial alignment. Choosing $\eta = 1/\lambda_{\max}$ gives

$$1 - \eta\lambda_{\min} \;=\; 1 - \frac{\lambda_{\min}}{\lambda_{\max}} \;=\; 1 - \frac{1}{\kappa(\boldsymbol{H}_p)},$$

hence

$$\|\boldsymbol{e}_t\|_2^2 \;\ge\; \left(1 - \frac{1}{\kappa(\boldsymbol{H}_p)}\right)^{2t} \|\boldsymbol{e}_0\|_2^2 \cos^2(\theta_0).$$

Finally, the stability condition $\eta \le 2/\lambda_{\max}$ ensures $|1 - \eta\lambda_k| \le 1$ for all $k$, so that each spectral component is non-expansive and the above bound is meaningful. If $\lambda_{\min}$ has multiplicity $r > 1$, the same argument applied to the corresponding eigenspace shows the projection of $\boldsymbol{e}_t$ onto that eigenspace decays as $(1 - \eta\lambda_{\min})^t$, and the stated bound holds with $\cos^2(\theta_0)$ interpreted as the squared cosine of the projection of $\boldsymbol{e}_0$ onto that eigenspace. □

# I VISUALIZING ATTENTION MAPS FOR VIT

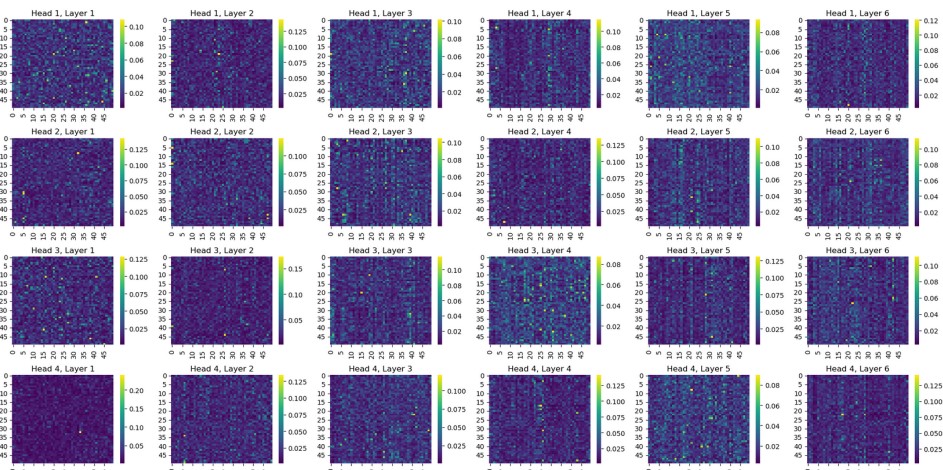

Figure 4: Attention maps obtained at the end of training of the model with $R \approx 20$.

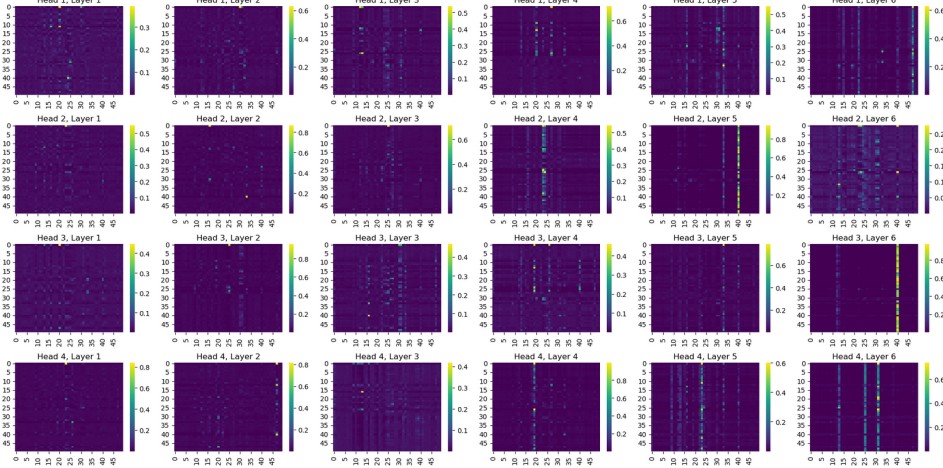

Figure 5: Attention maps obtained at the end of training of the model with $R \approx 10^7$.

**Attention sparsity in practice.** Sparse attention patterns are widely documented in trained transformers across different architectures and tasks. Shi et al. analyze attention importance in BERT and show that attention importance is highly concentrated on a small subset of token pairs, with many heads exhibiting very skewed importance distributions. Zhang et al. (2023) observe that large language models concentrate attention on a small set of "heavy hitter" tokens that account for most of the attention mass during inference. Vig & Belinkov (2019) analyze the structure of attention in a transformer language model and document systematic attention patterns, including heads that focus strongly on delimiter tokens and on nearby context. Zucchet et al. (2025) study how sparse attention emerges during training, showing that attention sparsity increases as models learn and that the timing of this emergence is influenced by data distribution and repetition. Zhai et al. (2023) demonstrate that attention entropy collapse (extremely concentrated, low-entropy attention) is a common training instability in transformers. Ji et al. (2021) provide a comprehensive analysis of attention value distributions, showing that attention activations are consistently sparse across layers and that this sparsity can be exploited for pruning and quantization at inference time. Taken together, these observations confirm that the sparse attention regime analyzed in Theorem 2 reflects conditions commonly encountered during transformer training and inference, rather than an artificial theoretical construct.

## J    ADDITIONAL EXPERIMENTS

### J.1    DYNAMICS UNDER JOINT TRAINING OF $W$, $p$, AND $v$

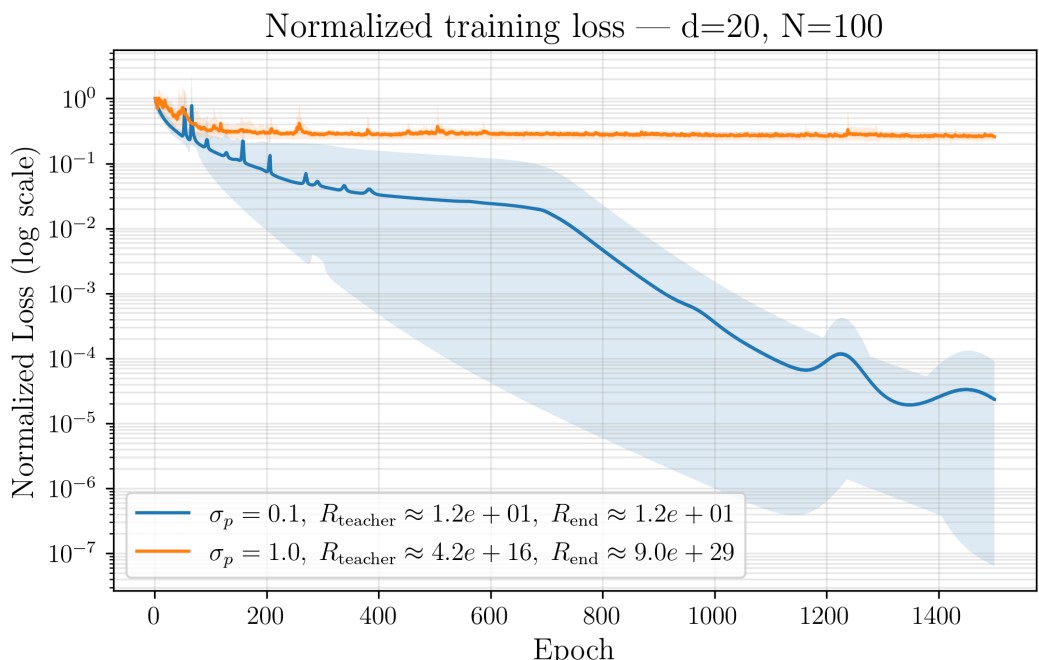

Figure 6: Training loss of GD under the joint training of $W$, $p$, and $v$. We do not observe a qualitative difference in the behavior of GD compared to when $W$ is frozen.

### J.2    ADDITIONAL SYNTHETIC RESULTS: CONVERGENCE VS. ATTENTION RATIO $R$

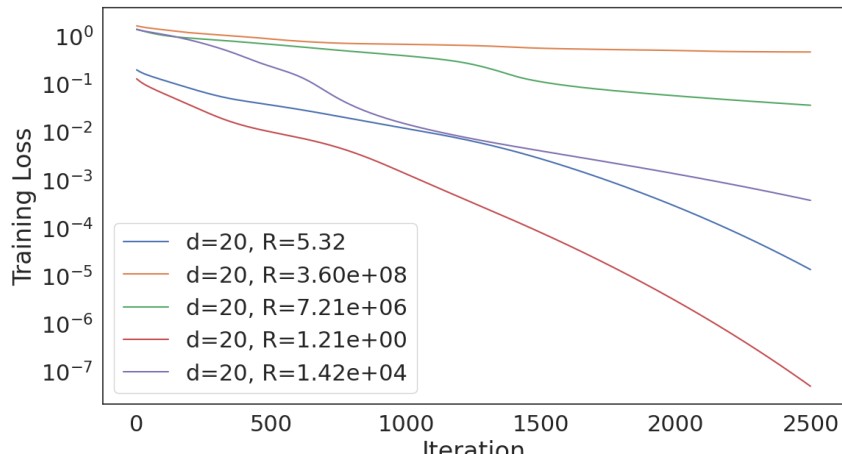

Figure 7: Training loss on synthetic data in the underparameterized regime (d=20, N=100). We sweep attention sparsity $R$ across five orders of magnitude. Consistent with Theorem 2, GD convergence degrades monotonically with R: nearly uniform attention ($R \approx 1$) yields rapid convergence to $10^{-7}$, while highly sparse attention ($R \approx 10^8$) causes stagnation above $10^{-7}$

## J.3 EMPIRICAL VALIDATION OF ASSUMPTION B

We empirically verify Assumption B on the trained Vision Transformer from Figure 2b. Our analysis spans all 6 layers and 4 attention heads, examining approximately 2000 test examples.

**Methodology.** For each layer $\ell$ and head $h$, we extract:

- Keys $K_i \in \mathbb{R}^{T \times D_k}$ for each example $i$
- Attention weights $a_i^* = \phi(K_i p^*)$ of the CLS token
- Value vectors $u_i = X_i v$

We define the important token set $S_i$ as tokens with attention weight $\geq 50\%$ of the maximum attention in that example.

**B1: Subspace Geometry.** We compute the signal subspace $I$ via PCA on the important keys $\{K_{i,S_i}\}$, retaining components explaining 90% of variance. The complementary subspace is $U = I^{\perp}$. For each head, we estimate the worst-case leakage ratio:

$$\rho_U = \max_{\|x_U\|=1, x_U \in U} \frac{\|K_{i,S_i} x_U\|}{\|K_{i,S_i^c} x_U\|}$$

by sampling 200 random unit vectors in $U$.

Table 1: Summary statistics for Assumption B1 across all layers and heads.

| Statistic | Value |
|---|---|
| Median $\rho_U$ | 0.244 |
| 90th percentile $\rho_U$ | 0.365 |
| Maximum $\rho_U$ | 0.431 |
| Fraction with $\rho_U < 1$ | 100% |

Table 1 shows that **all 24 layer-head combinations satisfy** $\rho_U < 1$, with median $\rho_U = 0.244$. This confirms that important keys lie predominantly in a lower-dimensional signal subspace, while unimportant keys span a complementary direction—precisely the structure required by (B1).

**B2: Value-Compatibility Margin.** For each example $i$ and head $h$, we compute:

$$\mu_i = \sum_j (a_i^*)_j (u_i)_j, \qquad \delta_i = \min_{j \in S_i} |(u_i)_j - \mu_i|_2$$

and report $\delta_{\min}$ across examples.

Table 2: Summary statistics for Assumption B2 across all layers and heads.

| Statistic | Value |
|---|---|
| Median $\delta_{\min}$ | 0.030 |
| 10th percentile $\delta_{\min}$ | 0.001 |
| Minimum $\delta_{\min}$ | 0.0002 |
| Fraction with $\delta_{\min} > 0$ | 100% |

Table 2 confirms that **all heads exhibit strictly positive value-compatibility margins**. While the margins are small in absolute terms, they are sufficient for the local analysis in Theorem 2, which requires only $\delta_i > 0$.

Table 3: Estimated condition numbers from attention sparsity.

| Statistic | Value |
|---|---|
| Median $\kappa(J^2)$ | $4.4 \times 10^5$ |
| 90th percentile $\kappa(J^2)$ | $2.1 \times 10^8$ |

**Condition Number Verification.** As a direct test of our theoretical predictions, we estimate the condition number contribution from the softmax Jacobian $J(a_i^*)$. By Lemma 1, $\lambda_{\min}(J(a)) \geq \min_j a_j$, so for sparse attention:

$$\kappa(J(a^*)^2) \approx \left( \frac{\max_j a_j}{\min_j a_j} \right)^2$$

The extremely large condition numbers in Table 3 directly validate Theorem 2's prediction that sparse attention induces severe ill-conditioning, explaining the observed slowdown of gradient descent.

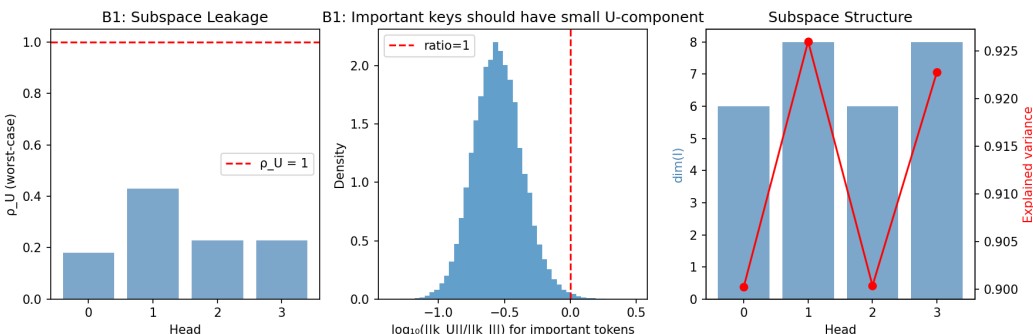

Figure 8: Visualization of Assumption B1 for the final transformer layer. **Left:** Worst-case leakage ratio $\rho_U$ across heads (all below 1). **Middle:** Distribution of $\log_{10}(\|k_U\|/\|k_I\|)$ for important tokens (concentrated below 0, confirming small U-component). **Right:** Dimension of signal subspace $I$ and explained variance.

## K WIKITEXT-2 EXPERIMENT DETAILS

**Dataset and preprocessing.** We use the WikiText-2-raw-v1 corpus from HuggingFace (`wikitext`, split `"wikitext-2-raw-v1"`). We apply the `distilbert-base-uncased` tokenizer from `transformers`, without an attention mask. All tokenized documents in the training split are concatenated into a single long sequence of token IDs. We then partition this sequence into non-overlapping chunks of fixed length $L = 64$: we discard the final partial chunk and reshape into a matrix of shape $(N, 64)$, where each row is one training example.

**Tasks.** We consider two tasks on these chunks.

- **Next-token prediction.** The input is the first 63 tokens of a chunk, and the target is the 64-th token. This is a standard language-modeling style next-token prediction problem.
- **Bag-of-words classification.** The input is the full 64-token sequence. The label is 1 iff the tokenized sequence contains the token "the" (according to the tokenizer's vocabulary), and 0 otherwise. This produces a simple binary classification problem that still requires aggregating information across the sequence.

**Model architecture.** For both tasks we use the same small transformer encoder, which we refer to as a "tiny transformer". The model has:

- token embeddings and positional embeddings with hidden size $d = 128$,

- a learned `[CLS]` token, which is prepended to the sequence,
- $L_{\text{layers}} = 2$ transformer blocks,
- $H = 4$ attention heads per block.

Each transformer block consists of a multi-head self-attention sublayer followed by a two-layer feed-forward network, both with residual connections and layer normalization. Concretely, we use `nn.MultiheadAttention` with `embed_dim = 128` and `num_heads = 4`, and a feed-forward network of the form

$$\text{FFN}(h) = W_2\, \phi(W_1 h + b_1) + b_2,$$

with width $4d$ and GELU nonlinearity. Positional embeddings are learned for all positions $0, \ldots, 64$ (one position for `[CLS]` plus 64 tokens). The final representation is the `[CLS]` vector from the last layer. For the bag-of-words task, this vector is passed to a linear classifier with 2 outputs; for next-token prediction, it is passed to a linear classifier with $V$ outputs, where $V$ is the tokenizer vocabulary size. We use standard cross-entropy loss in both cases.

**Training setup.** We train with mini-batch size 32, up to 10 epochs and at most 2000 gradient steps (we stop early if the step limit is reached). We run each configuration with 3 different random seeds. For each seed we set Python, NumPy, and PyTorch RNGs accordingly to ensure reproducibility. All experiments are run on a single GPU when available; otherwise, training defaults to CPU.

**Optimizers and hyperparameter search.** We compare SGD with momentum to Adam.

- **SGD.** We use SGD with momentum $0.9$. For each task (next-token prediction and bag-of-words) we perform a grid search over learning rates

$$\eta \in \{0.01, 0.05, 0.1, 0.5\}.$$

- **Adam.** We use Adam with weight decay fixed to $10^{-4}$. For each task we perform a grid search over learning rates

$$\eta \in \{10^{-4}, 5 \cdot 10^{-4}, 10^{-3}, 5 \cdot 10^{-3}\}.$$

For each optimizer–task pair and each candidate learning rate in the grid, we train 3 independent runs (different random seeds) with the same architecture and data. We then select the learning rate that gives the lowest mean final training loss across the 3 seeds.

**Metrics and plotting.** To study how attention evolves, we log both the training loss and a scalar measure of the "peakedness" of the `[CLS]` attention. At the beginning of training, we compute an initial loss $\ell_0$ on one mini-batch and an initial attention peakedness value. Every 50 steps, we log:

- the normalized loss $\ell_t/\ell_0$ on the current mini-batch, and
- an attention peakedness ratio: for each layer and head, we extract the attention distribution of the `[CLS]` token over sequence positions, compute the ratio between its maximum and minimum probabilities, then average this ratio over heads and over examples in the mini-batch. We finally average over layers to obtain a single scalar.

For each configuration, this yields a trajectory of normalized loss values and attention ratios over training. In the plots, we show the mean curve across the 3 random seeds and shade the region corresponding to one standard deviation around the mean. Thus, the shaded regions in Figure 3 represent $\pm 1$ standard deviation across seeds for both loss and attention peakedness.

