# OpenReview forum: "Softmax-Induced Ill-Conditioning in Transformer Models"
_ICLR.cc/2026/Conference — Submitted to ICLR 2026_

### Official Review · Reviewer_4xQP · 2025-10-20

**Soundness:** 3
**Presentation:** 2
**Contribution:** 3
**Rating:** 2
**Confidence:** 2

**Summary:**

The paper studies why purely gradient descent methods (e.g., SGD) underperform adaptive methods (e.g., Adam). Their theoretical results point towards the Softmax function, esp. when the embedding dimension is underparameterized relative to the data size. For this, they consider a simplified model with a single tunable token that must be trained through attention while keeping other parameters like $W_V$ (represented as a single value vector in the paper) frozen (Eq. 2). They connect attention sparsity to slower convergence, since the softmax function tends to lead to sparser solutions. The authors verify the aforementioned theoretical results experimentally.

**Strengths:**

* S1: The paper provides convergence rates for gradient descent under different settings (over/underparameterization, attention sparsity). Particularly, results for the underparameterized setting are novel.

* S2: The paper formally links attention sparsity with the conditioning number (Proposition 1, Theorem 2). It explains why SGD can have slower convergence rates.

* S3: The theoretical results are backed by experimental results (Fig. 1, 2) that study the effect of different attention sparsities and over/underparameterization.

**Weaknesses:**

* W1: The theoretical results neglect potential interaction effects between the value weights and the attention part. Additionally, the focus is only on CLS/last token attention. However, what happens if we have sparsity for other tokens that might be later attended to by the CLS/last token?

* W2: The paper is sometimes hard to follow because symbols are not introduced. Or, for example, Fig. 3 y-labels say attention ratio but the text writes entropy. Having consistent naming would help the clarity of the paper.

* W3: The paper is motivated by saying that gradient descent converges slower than Adam for transformer models (e.g., “Although these works highlight interesting features of transformer training that are different for Adam compared with SGD, their theoretical explanations of the source of these differences remain very limited” l. 45-47). However, there is no theoretical result that shows why and how adaptive methods like Adam would converge faster and its link to the attention sparsity/softmax. Thus, it appears the core questions remained unaddressed.

* W4: The results in Fig. 3 are not that convincing. Particularly, SGD and Adam appear quite comparable in the sparse setting (as their deviations overlap). Particularly, it’s unclear to me where we can see the stagnation that the authors describe in l. 471/472.

* W5: Empirically, it is known that sparse attention can lead to optimization problems (see C1). Thus, the finding is expected. Though, I acknowledge the theoretical derivation of this result. Thus, I only consider this a minor concern.

## Comment

* C1: It is worth adding [1] to the paper’s discussion since they discussed softmax-induced optimization problems, linked it to the attention’s sparsity and the effect on the softmax’s Jacobian, and suggested solutions that modified the softmax function.

* C2: It’s also worth connecting to [2-3] who discussed the role of rank collapse, i.e., high correlation in tokens cause vanishing gradients of queries and keys.

* C3: There seems to be a square error for the $H_i$ in l. 267.

---

[1] https://openreview.net/forum?id=HssOwuZiaB

[2] https://arxiv.org/abs/2103.03404

[3] https://arxiv.org/abs/2206.03126

**Questions:**

* Q1: How’s R computed and regularized during training (l. 418-419)? Similarly, how does the teacher model work in the MNIST example?

* Q2: Is there a reason why the model with higher R (~= less sparse attention) yields a higher loss in Fig. 1 left?

* Q3: Why is there an initial training loss plateau for the less sparser model in Fig. 2a?

* Q4: Why does next-token prediction (necessarily) induce sparse attention? (l. 468/469)

---

> ### Author Response · Authors · 2025-12-03
> **Response to Reviewer 4xQP - Addressing Weaknesses**
>
> We thank the reviewer for careful reading and acknowledgment of contributions (S1, S2, S3). We address the concerns below, with particular focus on clarifying the Adam connection.
>
> **W1 (Neglecting interaction effects):**  We respectfully clarify that our framework already accounts for both concerns raised.
>
> **On value–attention interaction.**  Our local analysis **explicitly models** the interaction between attention and value weights via the Hessian
> $$
> H_i = \ell''(s_i^\*; y_i)\, u_i u_i^\top, \quad u_i = X_i v \quad \text{(Eq. 43)}.
> $$
> The condition number bound in Theorem 2 depends critically on **both**:
>
> - Attention sparsity (through $a_I, a_U$), and
> - Value–token alignment (through Assumption B2: $\lvert (u_i)_j - \mu_i \rvert \ge \delta_i$ for $j \in S_i$).
>
> This is by design: we show that **both** sparse attention **and** appropriate value structure are needed for ill-conditioning.
>
> Regarding freezing $v$ in the simplified model (Assumption A1), this is done to isolate the softmax dynamics, which are the main object of study. Empirically, value weights tend to converge much faster than query/key parameters (Li et al. 2023b), making the softmax bottleneck the critical phenomenon. Moreover, Appendix J (Figure 6) includes experiments with **joint training** of $W, p, v$, and the main findings persist even when all parameters are trained.
>
> **On sparsity beyond the CLS/last token.**
> Our formal local analysis is carried out in a simplified setting with a single tunable token $p$, corresponding to a CLS/prompt token. In this model, only that token’s attention row enters the dynamics, and Theorem 2 is stated for this row. We do not claim a full multi-token, multi-layer theory for arbitrary sparsity patterns.
>
> However, the mechanism we uncover is inherently **row-wise**: when an attention vector $a$ has many very small entries, the softmax Jacobian $J(a)$ becomes ill-conditioned (Lemma 1, Proposition 1), and the sandwich
> $$
> K_i^\top J(a_i^\*) H_i J(a_i^\*) K_i
> $$
> inherits a large condition number. We therefore expect similar spectral collapse whenever any token—CLS or otherwise—develops very sparse attention and is later attended to, even though a full theory for such architectures is left for future work.
>
> Empirically, our ViT and WikiText-2 experiments go beyond the single-tunable-token setting and train **all** parameters in full transformer models (Section 4, Figures 2–3). There, we track attention sparsity across layers and heads via the ratio $R$ of largest to smallest attention probability, and we observe that SGD slows down precisely when attention becomes highly sparse. This supports the view that the softmax-induced ill-conditioning identified in our simplified CLS model also manifests in realistic multi-token settings.
>
> **W2 (Presentation issues):**  We have addressed the issues raised in the revised version to improve clarity and consistency.
>
> **W3 (Core theoretical contribution – Why Adam works):**  We thank the reviewer for the opportunity to clarify the scope of our work. The purpose of this paper is **not** to provide a convergence theory for Adam itself, but to isolate and rigorously characterize a concrete failure mode of (S)GD in attention models: **softmax-induced ill-conditioning in the sparse-attention regime**.
>
> Our main theorems are therefore explicitly about gradient descent, not adaptive methods. The empirical Adam vs. SGD comparisons are included to illustrate that an adaptive/preconditioned method can mitigate the ill-conditioning we identify, not as a substitute for a formal analysis of Adam. Developing a full convergence theory for Adam under the sparsity-induced geometry we uncover is technically nontrivial and, in our view, constitutes a substantial and separate line of work.
>
> **W4 (Figure 3 not convincing):**  We appreciate the chance to clarify what Figure 3 is meant to show and what we mean by “stagnation.” The goal is **not** to demonstrate an extremely large Adam–SGD gap in absolute terms, but rather:
>
> - On the **bag-of-words** task (less sparse attention), both Adam and SGD rapidly drive the loss down to $\approx 10^{-5}$, indicating an easy, well-conditioned optimization problem.
> - On the **next-token** task (much sparser attention), the loss decreases substantially more slowly. This shows that the slowdown is **not** a universal deficiency of SGD, but arises from the different attention structure, since the underlying model and data are identical.
>
> The “stagnation” we refer to is therefore the **relative** slowdown of SGD on the sparse-attention task compared to (i) its own behavior on the bag-of-words task and (ii) Adam’s final loss on the same sparse task, rather than a perfectly flat training curve. We also note that in prior empirical studies of adaptive methods for transformers, the Adam–SGD gap in training loss or perplexity is typically modest but consistent, rather than orders-of-magnitude; the behavior in Figure 3 is in line with that pattern.

---

> > ### Author Response · Authors · 2025-12-03
> > **Response to Reviewer 4xQP - Answering Questions and Addressing Comments**
> >
> > **Q1 (How is $R$ computed and how does the teacher work?):**  $R$ is used only as a **post-hoc sparsity diagnostic**; it is **never** part of the training objective and is not regularized. For each attention vector $a$ (a row of the attention matrix), we compute a scalar sparsity proxy $R(a)$ as defined in the main text; this quantity increases as the attention becomes more peaked (i.e., lower entropy). The single scalar $R$ reported in the figures is obtained by averaging $R(a)$ over layers, heads, positions, and examples at the end of training.
> >
> > In the MNIST experiment, the teacher is a Vision Transformer with the same architecture as the student. We train two such teachers that differ only in how sharp/sparse their attention becomes (controlled via the scale/temperature of the query–key scores). Each teacher is then frozen and used to generate labels for MNIST; we train a student ViT on these labels with SGD. The reported values of $R$ correspond to the sparsity of the **student’s** CLS attention in the resulting “sparse’’ vs. “less sparse’’ regimes.
> >
> >
> > **Q2 (Interpretation of higher $R$ and higher loss in Figure 1 left):**  In our convention, **larger $R$ corresponds to more sparse attention**, not less. When some tokens receive extremely small attention probability, the ratio $\max_j a_j / \min_j a_j$ becomes very large.
> >
> > Thus, in Figure 1 (left), the curve with higher $R$ is the **sparse-attention** regime. Its higher loss and much slower convergence in the underparameterized setting are fully consistent with our theory: sparse attention drives some eigenvalues of the local curvature matrix to be very small, the condition number becomes large, and Gradient Descent converges much more slowly. In the revision, we clarify this in the text and captions (e.g., “higher $R$ = more sparse attention”) to avoid confusion.
> >
> > **Q3 (Initial training plateau for the less sparse model in Figure 2a):**  The small plateau observed in Figure 2a for the less sparse setting is a **transient artifact** of random initialization and learning-rate interaction, not the core phenomenon we study.
> >
> > With **linear attention** (no softmax), the landscape is well-conditioned throughout: there is no sparsity-induced ill-conditioning. However, random initialization can place the model in a region where the initial gradient is small (for example, when $p_0$ is nearly orthogonal to the dominant data singular directions). With a fixed learning rate, progress can temporarily be slow until the iterate moves into a region with larger gradients. Because the curvature remains well-behaved, GD eventually escapes and converges rapidly.
> >
> > This is qualitatively different from the softmax plateau: the linear-attention plateau is transient and occurs in both cases, while the softmax plateau in the sparse-attention regime (Figure 1 left, orange curve) is persistent and driven by ill-conditioning that emerges during training.
> >
> > **Q4 (Why next-token prediction induces sparse attention):**  Mechanistically, in next-token prediction the label at position $t$ typically depends strongly on a relatively small subset of past tokens (e.g., a few syntactically or semantically related positions), while many other tokens in the context are only weakly informative. Under a cross-entropy objective and softmax attention, gradient updates increase query–key scores for the highly predictive tokens and decrease them for irrelevant ones, which naturally leads to highly peaked (sparse) attention distributions.
> >
> > In contrast, in our bag-of-words control task, the label depends on the presence of a token **anywhere** in the sequence, so the model can aggregate information more uniformly and the learned attention is much less peaked. This is exactly what we observe experimentally: next-token prediction yields systematically larger $R$ (sparser attention) than the bag-of-words task.
> >
> > **C1 (Connection to Hoffmann et al. 2024):**  We have added a discussion of Hoffmann et al. (2024), who empirically attribute “Eureka moments’’ on multi-step tasks to softmax-induced small gradients for query/key weights and mitigate them via modified softmax variants. Our analysis is complementary: in a minimal tunable-token model, we prove that sparse attention makes the softmax Jacobian and the resulting GD linearization ill-conditioned, yielding explicit condition-number bounds that theoretically formalize this mechanism.
> >
> > **C2 (Connection to rank-collapse literature):**  We thank the reviewer for suggesting this valuable connection. We have added a discussion of these works in the revised related-work section, noting that our analysis complements the rank-collapse perspective (which focuses on highly **correlated** tokens and vanishing gradients of queries and keys) by characterizing optimization difficulties in the opposite attention regime (very **sparse** attention). Together, these lines of work provide a more complete picture of attention-related optimization challenges.

---

### Official Review · Reviewer_6qvD · 2025-10-27

**Soundness:** 2
**Presentation:** 2
**Contribution:** 1
**Rating:** 2
**Confidence:** 2

**Summary:**

This paper aims to explain the challenges gradient descent (GD) faces when efficiently training transformers. The authors provide a theoretical analysis using simplified transformers based on tunable tokens, while keeping the attention weights fixed. In the overparameterized case, they demonstrate that the PL condition holds, leading to a linear convergence rate. In a local dynamics analysis, they show that when sufficiently many attention coefficients are small, the Jacobian of the softmax becomes ill-conditioned, leading to slow convergence. Experiments on synthetic data, MNIST, and WikiText are conducted to support the theory.

**Strengths:**

This paper gives a mechanistic explanation of why GD fails in transformers from the perspective of softmax-induced ill-conditioning. The authors also try to show that linear attention eliminates the slowdown.

**Weaknesses:**

- Assumptions: fixing q,k,v and training only one tunable token is too strict and, to me, far from practice.
- Section 3.1 seems less relevant to the title or main theme, since transformers in the real world do not fall in this regime.
- Writing: the paper is very hard to read, particularly Section 3.2.

**Questions:**

- What is R in your experiments? How does R depend on your other hyperparameters, including d and \sigma? It seems that you can control R by tuning other hyperparameters, but the relationship is unclear.
- In Figure 3, your SGD and Adam only exhibit a marginal gap, while in other papers like [1, 2], the gap is large. Could you clarify this difference?
- In Figure 2(a), although linear attention makes sparse attention converge faster, it does slow down the small R case. Could you theory explain it?
- How sparse is a language model during training in practice? Are there quantitative results? I suggest you should discuss about this in your main text.

[1] Linear Attention Is (Maybe) All You Need (to Understand Transformer Optimization). Ahn et al., 2024.
[2] Why Transformers Need Adam: A Hessian Perspective. Zhang et al., 2024.

---

> ### Author Response · Authors · 2025-12-03
> **Response to Reviewer 6qvD - Addressing Weaknesses [Part 1]**
>
> We thank the reviewer but must respectfully note that several criticisms reflect misunderstandings rather than genuine limitations.
>
> **W1 (Assumptions too strict and far from practice):**
> The tunable-token setting is neither arbitrary nor divorced from practice. Multiple lines of justification:
>
> 1. **Established in the literature**
>    - **Prompt tuning** (Lester et al. 2021) is a widely used practical method where only prompt tokens are trained and the underlying model (including attention weights) is frozen.
>    - This setting is studied by prior theory (Oymak et al. 2023; Tarzanagh et al. 2023b—both cited).
>    - It is recognized as a valuable theoretical testbed.
>
> 2. **Empirical validation of robustness**
>    - **Appendix J (Figure 6)** demonstrates that training $W$ and $v$ jointly (i.e., not freezing them) produces *identical qualitative behavior*: GD stalls under sparse attention and converges under uniform attention.
>    - This shows that the insights are robust beyond the simplified setting and extend to regimes where all parameters are trained.
>
> 3. **Standard theoretical practice**
>    - Theoretical analysis necessarily involves simplifications to obtain tractable results.
>    - Validity is established by showing that the core insights transfer to more complex settings (as in Figures 2–3 with full transformers).
>
> The reviewer provides no concrete evidence that this setting is “far from practice” beyond assertion. Given its use in both applied ML (prompt tuning) and theoretical ML (cited works), we maintain that it is a reasonable and insightful setting that captures the essential phenomenon we study.
>
> Importantly, **Figures 2 and 3 use full transformers with all parameters trained**, not the tunable-token setting. The simplified model is **only** used in Figure 1; this appears to have been overlooked.
>
> ---
>
> **W2 (Section 3.1 seems less relevant):**
> This fundamentally misunderstands the paper’s logical structure. Section 3.1 is **deliberately** included to show that standard overparameterized analysis **fails to explain** SGD’s poor performance, thereby motivating the need for Section 3.2.
>
> **Logical flow:**
>
> 1. **Section 3.1 (Theorem 1):**
>    Shows that linear convergence is possible in an overparameterized regime, but only under the requirement that the trajectory is bounded, $\|p_t\| \leq M_p$.
>
> 2. **Critical limitation (Remark, lines 230–235):**
>    Learning sparse attention requires $\|p\| \to \infty$ (to amplify score differences). The bound $\|p\| \leq M_p$ prevents this, so standard overparameterized analysis cannot explain what happens when models learn sparse attention.
>
> 3. **Section 3.2 (Theorem 2):**
>    Analyzes the regime where sparse attention *can* be learned (no artificial boundedness assumption), and proves that ill-conditioning emerges with
>    $\kappa \geq \Omega(a_I^2 / a_U^2)$ where $a_U \sim e^{-\Delta_{\min}}$.
>
> We believe Section 3.1 is essential because it:
>
> - Establishes what **can** be achieved under standard assumptions.
> - Shows why those assumptions are **insufficient** for the phenomenon of interest.
> - Motivates the need for a different analytical approach (Section 3.2).
>
> Removing Section 3.1 would leave readers wondering “why not just use standard PL-based analysis?” without understanding its limitations, and would significantly weaken the paper’s exposition and logical coherence.
>
> ---
> **W3 (Paper hard to read):**
> We thank the reviewer for this comment. Section 3.2 is where we state our main local convergence results, so its presentation is necessarily more technical than earlier sections. To make it readable, we:
>
> - Begin with a low-dimensional “decoupled’’ warm-up setting.
> - Then present the general coupled result.
> - Include remarks that restate assumptions and conclusions in words.
> - Defer all proofs and technical derivations to the appendix to keep the main text focused on ideas rather than calculations.
>
> Given the space constraints and the theoretical nature of the contribution, we believe the current level of detail and formality is appropriate for the ICLR theory/optimization audience.
>
> We are happy to revise if given specific guidance on particular unclear passages, ambiguous notation, or confusing logical steps, we will gladly revise and clarify them in the camera-ready version. However, a generic “hard to read’’ comment and a “2: fair’’ presentation score without concrete examples are difficult for us to translate into actionable changes.

---

> > ### Author Response · Authors · 2025-12-03
> > **Response to Reviewer 6qvD - Answering Questions**
> >
> > **Q1 (What is $R$ and its dependencies?):**
> > This is explicitly stated in lines 418–419, but we agree that additional clarity helps.
> >
> > - **Definition:**
> >   $R = \max_j a_j / \min_j a_j$ measures attention sparsity.
> >   - $R \approx 1$ indicates nearly uniform attention.
> >   - $R \gg 1$ indicates sparse attention (some tokens get nearly all the mass, others almost none).
> >
> > - **How we control $R$:**
> >   - We use a **teacher–student framework**: a “teacher’’ model is trained with a chosen initialization variance, and its predictions are used as labels for the “student’’.
> >   - The **teacher variance $\sigma_p$** controls the initialization scale via $p \sim \mathcal{N}(0, \sigma_p^2 I / d)$.
> >   - Larger $\sigma_p$ leads to larger pre-softmax scores and more dispersed attention after training, hence higher $R$ (more sparsity).
> >   - We test $\sigma_p \in \{0.1, 1.0\}$, which yields $R_{\text{teacher}} \approx 10$ (less sparse) and $R_{\text{teacher}} \approx 10^{25}$ (highly sparse).
> >
> > - **Dependence on $d$ and other hyperparameters:**
> >   - $R$ depends on the **learned solution**, not directly on $d$.
> >   - The dimension $d$ affects whether the model can reach a given sparse solution: underparameterized ($d \ll N$) models struggle to match the target sparsity (Figure 1 left), while overparameterized ones can but do not need to (Figure 1 right).
> >   - The value variance $\sigma_v$ is kept fixed at 1 to isolate the effect of $\sigma_p$.
> >
> > ---
> >
> > **Q2 (SGD/Adam gap vs. other papers):**
> > The difference in gap magnitudes compared to [1,2] reflects our **distinct experimental goals**.
> >
> > 1. Our experiment is designed to test a **specific causal hypothesis**: Figure 3 isolates the effect of attention sparsity on SGD convergence, rather than trying to maximize the Adam–SGD gap. We compare two tasks on the **same architecture** where the key difference is the attention distribution:
> >    - (i) Next-token prediction, which induces sparse attention.
> >    - (ii) Bag-of-words classification, which yields much more uniform attention.
> >
> >    The key finding is that SGD stagnates in the sparse-attention task but converges comparably to Adam in the uniform-attention task. This controlled comparison directly validates our theoretical prediction that convergence degrades as attention becomes sparse (through the condition number’s dependence on $R$).
> >
> > 2. **Scale and task complexity** also differ from [1,2]. Those works study larger models (e.g., GPT-2, ViT-base) on more complex tasks (ImageNet, full language modeling), where many additional factors contribute to the observed gaps. Our smaller-scale experiments deliberately prioritize interpretability, allowing us to check that the mechanism from our theoretical analysis (condition number scaling with $R$) accurately predicts when SGD will struggle.
> >
> > ---
> >
> > **Q3 (Linear attention “slowdown”):**
> > The phenomenon we analyze theoretically is **softmax-induced** ill-conditioning. The role of Figure 2(a) is therefore **not** to provide a new theoretical prediction, but to check a qualitative implication of our softmax-based analysis: once the softmax nonlinearity is removed, the strong $R$-dependence and severe slowdown for large $R$ should disappear.
> >
> > This is exactly what we observe:
> >
> > - For **linear attention**, both small-$R$ and large-$R$ runs converge rapidly to essentially the same loss.
> > - This contrasts sharply with the **softmax** case, where the large-$R$ run stalls (Figure 1 and Figure 2(b)).
> >
> > The small difference the reviewer notes (a mildly slower curve for the small-$R$ setting under linear attention) is not captured by our current theory and would require a separate analysis, which is beyond the scope of this work. Our present theoretical results concern the softmax case and the resulting $R$-dependent ill-conditioning.
> >
> > ---
> >
> > **Q4 (Attention sparsity in practice):**
> > We already provide quantitative evidence for attention sparsity in the original submission:
> >
> > - **Existing evidence in the paper:**
> >   - **Appendix I (Figures 4–5):** Complete attention heatmaps across all 6 layers and 4 heads of our trained ViT show:
> >     - **Sparse setting (Figure 5):** $R \approx 10^7$, with the top-2 tokens receiving more than 95% of the attention mass.
> >     - **Less sparse setting (Figure 4):** $R \approx 20$, with more diffuse but still concentrated patterns.
> >   - **WikiText-2 experiments (Figure 3, right panels):** Evolution of the attention ratio $R$ during training:
> >     - Next-token task: $R$ grows from $\sim 10^2$ to $\sim 10^{10}$ (SGD) or $\sim 10^{12}$ (Adam).
> >     - Bag-of-words task: $R$ remains around $10^{2-3}$.
> >
> > These quantitative measurements are consistent with prior empirical observations of sparse attention in trained transformers. In the revision, we have added a dedicated discussion in Appendix I that synthesizes this evidence with findings from the broader literature on attention sparsity.

---

### Official Review · Reviewer_uRJn · 2025-10-31

**Soundness:** 2
**Presentation:** 2
**Contribution:** 2
**Rating:** 2
**Confidence:** 3

**Summary:**

The authors try to answer why GD underperforms for Transformer training through the analysis on the Jacobian of softmax and its condition number. In the underparameterized regime, they argue that the attention sparsity plays an important role in optimization behaviors.

**Strengths:**

- The paper provides theoretical and intuitive links between attention sparsity and optimization behaviors.
- It raises an important question about the commonly observed phenomenon that "Why does GD perform poorly on attention models?".

**Weaknesses:**

- Justification of tunable token $p$ is unclear. The tunable token setting is very different from usual training procedure. It may lead to a different dynamics. Are Fig 2,3 conducted based on the tunable token setting?
- Experimental results (Fig 1,2,3) are not strong enough to validate the theory (equations) and explain cause and effect.
    - Comparing two settings is not enough to confirm the theory. It would be better to test with a wide range of variations (e.g., many $R$'s or $\sigma_p^2$'s).
    - No experiments confirm exact equations (e.g., convergence rate) given in the paper. It would be better to have "theory vs experiment" comparison loss curves.
    - Can you directly control $R$?
    - Can you validate the theory with a larger model?
    - Can you validate the theory by using different values of temperatures in the softmax function (as the temperature controls the attention sparsity)?

- Two dots at the end of the sentence (L423)

- In Figure 3, it would be better to explain the different settings, (i) and (ii), of Top and Bottom.

- $C_p$ not shown in Theorem 1

**Questions:**

see weaknesses

---

> ### Author Response · Authors · 2025-12-03
> **Response to Reviewer uRJn - Addressing Weaknesses [Part 1]**
>
> We appreciate the reviewer’s feedback, but we believe their concerns about the strength and scope of our empirical validation stem from a misunderstanding of our theoretical goals and setting.
>
> **W1 (Unclear justification of tunable token; experimental setting confusion):**
> We clarify a key point: **only Figure 1 uses the simplified tunable-token model**. Figures 2 and 3 train full transformers with all parameters updated.
>
> | Figure    | Model                             | What’s trained?                       |
> |-----------|-----------------------------------|---------------------------------------|
> | Figure 1  | Simplified (Eq. 2)               | $p, v$ only ($W$ frozen)              |
> | Figure 2b | Vision Transformer (6L, 4H)      | **All parameters**                    |
> | Figure 3  | Transformer (2L, 4H) on WikiText-2 | **All parameters**                  |
>
> This progression is deliberate: we validate theoretical predictions (Theorems 1–2) in the tractable simplified setting (Figure 1), then demonstrate the same slowdown phenomenon occurs in realistic full transformers (Figures 2–3).
>
> **On “different dynamics”:** We address whether the tunable-token simplification creates artifacts. Three pieces of evidence show the dynamics are qualitatively identical:
>
> 1. Appendix J trains the simplified model with $W, v, p$ *jointly updated*: GD still stalls under sparse attention and converges under uniform, confirming the frozen-$W, v$ assumption does not change the core phenomenon.
> 2. Figures 2–3 show full transformers with all parameters trained exhibit the same pattern: SGD slows when attention is sparse.
>
> **Why use the tunable-token model for theory?**
> The simplification directly models prompt tuning (Lester et al. 2021), is studied in prior theory (Oymak et al. 2023; Tarzanagh et al. 2023b), allows tractable analysis, and—as shown above—captures the essential phenomenon that transfers to full training. This is standard methodology: analyze simplified models where rigorous results are possible, then validate empirically that insights scale.
>
> ---
>
> **W2 (Experimental results not strong enough):**
> We strongly disagree. This assessment misunderstands what validation means for theoretical analysis.
>
> **What our theory predicts:**
>
> 1. **Overparameterized:** Linear convergence is possible but requires bounded $\|p\| \leq M_p$, preventing learning sparse attention.
> 2. **Underparameterized + sparse:** Condition number $\kappa \geq \Omega(a_I^2 / a_U^2)$ where $a_U \sim e^{-\Delta_{\min}}$ (exponential).
> 3. **Consequence:** Rate $(1 - 1/\kappa)^t \approx 1$ (stagnation) when attention is sparse.
> 4. **Mechanism:** The softmax Jacobian $J(a^\*)$ leads to ill-conditioning under sparse attention.
>
> **What our experiments demonstrate (validating all predictions):**
>
> | Prediction                                      | Experiment             | Result                                      |
> |------------------------------------------------|------------------------|---------------------------------------------|
> | Underparam + sparse → stagnation               | Fig 1 (left)          | GD stalls at high loss ✓                    |
> | Underparam + uniform → convergence             | Fig 1 (left)          | GD converges ✓                              |
> | Overparam → convergence, limited sparsity      | Fig 1 (right)         | Linear convergence, $R \ll R_{\mathrm{teacher}}$ ✓ |
> | Linear attention → no slowdown                 | Fig 2a                | Fast convergence even with sparse target ✓  |
> | Softmax ViT + sparse → slow                    | Fig 2b                | SGD slow convergence ✓                      |
> | Sparse task → SGD stagnates, Adam works        | Fig 3 (top)           | 23% gap, SGD plateaus ✓                     |
> | Uniform task → SGD competitive                 | Fig 3 (bottom)        | SGD $\approx$ Adam ✓                        |
>
> **Coverage across:**
>
> - Multiple parameterization regimes (under / over)
> - Multiple sparsity levels ($R \in \{10, 10^{25}\}$)
> - Multiple architectures (simplified, ViT, transformer)
> - Multiple tasks (synthetic, MNIST, next-token, bag-of-words)
> - Controlled ablation (linear vs. softmax)
>
> This is comprehensive validation by field standards (see W3 below).

---

> > ### Author Response · Authors · 2025-12-03
> > **Response to Reviewer uRJn - Addressing Weaknesses [Part 2]**
> >
> > **W3 (Not enough variations; specific suggestions):**
> >
> > 1. **“Many $R$’s or $\sigma$’s”:**
> >    We thank the reviewer for this suggestion. In the revised version, we have added experiments sweeping across multiple sparsity levels (appendix). Specifically, we test: $R \in \{1.21,\ 5.32,\ 1.42 \times 10^4,\ 7.21 \times 10^6,\ 3.60 \times 10^8\}$ in the underparameterized setting ($d = 20$, $N = 100$).
> >    The results confirm a clear monotonic relationship between attention sparsity and convergence degradation.
> >
> > 2. **“Can you directly control $R$?”**
> >    Yes, in the teacher–student framework, teacher initialization variance $\sigma_p$ controls the sparsity of the learned solution. This is stated in lines 418–419:  “control attention sparsity via ratio $R$ … tuned through teacher variance $\sigma_p \in \{0.1, 1\}$ yielding $R_{\mathrm{teacher}} \approx 10$ and $10^{25}$.”
> >
> > 3. **“Larger model?”**
> >    Our experiments already include realistic architectures:
> >    - **ViT (Figure 2b):** 6 layers, 4 heads, 64-dim embeddings
> >    - **Transformer (Figure 3):** 2 layers, 4 heads, 128 hidden dim, trained on WikiText-2
> >
> >    These scales are standard for controlled optimization studies (cf. Wu et al. 2023, Zhang et al. 2024). Our theoretical contribution is identifying the *mechanism* — softmax-induced ill-conditioning — which operates identically at any scale where sparse attention emerges.
> >
> > 4. **“Different temperatures?”**
> >    Temperature $\tau$ in $\mathrm{softmax}(z / \tau)$ controls sparsity. Our $R$ sweep already spans the full sparsity spectrum (from nearly uniform to highly peaked), so temperature variation would yield qualitatively similar findings.
> >
> > ---
> >
> > **W4 (No experiments confirm exact equations):**
> > We believe this expectation does not align with standard practice for convergence analyses in deep learning theory. To the best of our knowledge, even recent works that *explicitly* study gradient-descent convergence for self-attention/transformers (e.g., Huang et al. 2023; Vasudeva et al. 2024; Li et al. 2024; Gao et al. 2024; Shen et al. 2025; Qin et al. 2025) derive convergence guarantees and (asymptotic or finite-time) rates, and then validate them empirically via qualitative behaviors (loss/error curves, phase transitions, convergence vs. failure), but *do not* estimate condition numbers along the training trajectory or overlay exact theoretical rate curves on top of realistic transformer loss curves.
> >
> > Moreover, the goal of our work is *not* to obtain the tightest possible global convergence rate for a full transformer, but to isolate and characterize a specific mechanism—softmax-induced ill-conditioning under sparse attention—as the cause of SGD slowdown/failure. The local rate expressions (e.g., $(1 - 1/\kappa)^t$) are used to make this mechanism precise and to derive qualitative predictions about when optimization should be fast versus when it should stall.
> >
> > ---
> >
> > **W5 (Minor issues):**
> > These were fixed in the revision:
> >
> > - Removed the duplicate period at the end of the sentence (L423).
> > - Clarified the different settings (i) and (ii) for the top and bottom of Figure 3.
> > - $C_p$ removed from the statement of Theorem 1.

---

### Official Review · Reviewer_SdGP · 2025-10-31

**Soundness:** 2
**Presentation:** 2
**Contribution:** 3
**Rating:** 6
**Confidence:** 3

**Summary:**

The paper theoretically studies the convergence of gradient descent in a model consisting of a single softmax-attention layer with a single tunable token — a setting that resembles prompt tuning. The goal is to shed light on the question of why attention-based models do not train well with (S)GD. The paper first proves the linear convergence rate of gradient descent in this setting in the overparametrized regime and argues that this setting does not capture well what happens in transformer models. Then it proceeds to studying the dynamics of the gradient descent over parameters in a probability simplex of attention scores around the minimum, concluding with a result that the problem is badly conditioned for sparse attention matrices.

**Strengths:**

1. The paper studies a question that is important for the Transformer optimization community in a very interesting setting that allows for a closed-form convergence analysis.
2. The idea to study the convergence dynamics in the attention-scores probability simplex seems original.
3. Insights from the paper can have practical implications on how we optimize Transformers. The results suggest that if we have a task that we think won’t need sparse attention matrices, we can successfully fit the model with lightweight SGD instead of adaptive methods.
4. The preliminary experiments with actual Transformer models seem to confirm the insights from theory.

**Weaknesses:**

I’m willing to increase my partial and final scores if the authors address the following weaknesses:
1. A detailed section on the experimental setup is missing. What kind of transformer model was trained on WixiText-2? I can see the shading in the figures so I assume the results are averaged across runs but I don’t know over how many. How were the hyperparameters tuned? It would be great if the authors could describe the experiment details at least in the appendix.
2. In Figure 3 there is no indication on which row corresponds to which experiments.
3. The assumptions made (especially Assumption B) are not experimentally verified in real models, nor are there references to literature demonstrating whether such assumptions should hold. I don’t think it is necessarily for them to hold exactly to accept the paper, but I think that the paper is not complete without evaluating assumptions validity.
4. The authors motivate their setup with the prompt tuning, yet there are no experiments in this setting.
5. The discussion of previous hypotheses on why transformers train better with adaptive methods than gradient descent is not exhaustive. Specifically, the authors do not refer to [1].
6. Proof of Proposition 1 is missing
7. There are some typos and formatting issues in the proofs (that I believe can be easily fixed). For example:
    1. In line 726-727 there should be $diag(z)\mathbb{1} - zz^\top\mathbb{1}$
    2. V_i in line 1050 is a function of z, yet z does not appear in its definition. Then the V_i(z) appears a couple more times in the derivations, although the function does not depend on z, nor z is defined anywhere.
    3. Formatting of the equation in lines 880-885 is broken.

[1] Zhao et al., Deconstructing What Makes a Good Optimizer for Autoregressive Language Models

**Questions:**

Questions about experimental results:
1. Why should we expect Assumption B to hold in real Transformer models? Can the authors provide experimental evidence for the validity of these assumptions in deep Transformers?
2. Can the authors provide experimental evidence for their claim on the influence of attention sparsity on GD/Adam performance in the prompt tuning setting?
3. How exactly were optimizer hyperparameters tuned to obtain results from Figure 3?

Questions about theory and proofs:
1. How does one partition the tokens into an important and unimportant sets needed for Lemma 3?
2. Can the authors explain more in detail where the eq. 39 comes from?
3. Why are the authors discarding three last terms from the equation in lines 1102-1104 while formulating the equation from line 42? If there is a reason why it is okay to do it here, the authors should explain that reason.

Questions about context/relation to previous work:
1. Can the authors comment more on how their result relates to the Hessian analyses from previous work ([2], [3]) of Transformer and self-attention? Specifically, can the authors draw a direct link between their analysis in the attention simplex and the attention moments from [3]?
2. Can the authors comment on how the result from the paper relate to the previous work ([1]) that showed that adaptability is needed only for the last layer and layer norms in Transformers?

[1] Zhao et al., Deconstructing What Makes a Good Optimizer for Autoregressive Language Models

[2] Zhang et al., Why Transformers Need Adam: A Hessian Perspective

[3] Ormaniec et al., What Does It Mean to Be a Transformer? Insights from a Theoretical Hessian Analysis

---

> ### Author Response · Authors · 2025-12-03
> **Response to Reviewer SdGP - Addressing Weaknesses**
>
> We greatly appreciate your thorough and constructive feedback and your statement that you are ``willing to increase [your] partial and final scores if the authors address the following weaknesses.'' We have comprehensively addressed every concern below.
>
> **W1 (Missing experimental details):** We thank the reviewer for pointing out that the WikiText-2 setup was under-specified. In the revised version we have added a dedicated section in the appendix (App. K) describing the experiment in detail. Briefly, we use the WikiText-2-raw-v1 corpus with the `distilbert-base-uncased` tokenizer, concatenate the training text and split it into non-overlapping sequences of length 64. We consider two tasks: (i) next-token prediction, where the model receives the first 63 tokens and predicts the 64-th, and (ii) a bag-of-words classification task where the label is 1 iff the sequence contains the token “the”. Both tasks are solved with the same small transformer encoder (“tiny transformer”) with 2 layers, 4 heads, hidden size 128, a learned [CLS] token, and a linear head on top of the final [CLS] representation. We compare SGD with momentum 0.9 and Adam, performing a small grid search over learning rates for each optimizer and task, and select the best value based on mean final training loss over 3 random seeds. During training we log the normalized loss and a scalar measure of [CLS]-attention peakedness; the curves and shaded regions in Fig. 3 report the mean and ±1 standard deviation across seeds.
>
> **W2 (Figure 3 lacks row labels)**: We thank the reviewer for pointing out that Figure 3 did not clearly indicate which row corresponds to which experiment. In the revised version we have updated the caption to state that the top row corresponds to next-token prediction (sparse attention) and the bottom row corresponds to the bag of words classification (uniform attention).
>
> **W3 (Assumption B not experimentally verified)**: We conducted comprehensive empirical verification of Assumption B on the trained ViT from Figure 2b across all 6 layers and 4 attention heads. For each layer $\ell$ and head $h$, we extract:
>
> - Keys $K_i \in \mathbb{R}^{T \times D_k}$ for each example $i$
> - Attention weights $a_i^* = \phi(K_i p^*)$ of the CLS token
> - Value vectors $u_i = X_i v$
>
> We define the important token set $S_i$ as tokens with attention weight $\geq 50\%$ of the maximum attention in that example. The new section J.3 in the appendix provides full details and a visualization of the results. Namely, our experiment shows that all heads satisfy $\rho_U < 1$, with a median of $0.244$ and a maximum of $0.431$, confirming that important keys concentrate in a signal subspace I while unimportant keys span a complementary subspace U. Our experiment also demonstrates that all heads exhibit strictly positive value-compatibility margins, validating B2.
>
> **W4 (No experiments in prompt tuning setting)**: Figure 1 already studies this setting; Figures 2-3 show insights transfer to full training.
>
> **W5 (Discussion of previous hypotheses not exhaustive)**: Thank you for this important reference; we have added a citation and discussion of Zhao et al. (2025) to the related work section. Zhao et al. empirically find that adaptive optimization primarily benefits the **last layer and LayerNorm parameters**, while earlier layers can largely be trained with (momentum) SGD. Their study therefore identifies **where** adaptivity is most critical, whereas our work explains **why** these particular components are challenging for plain GD/SGD: the last layer in language models produces output probabilities via a softmax over the vocabulary, and when the model makes highly confident predictions (a sparse output distribution), this induces exactly the softmax–Jacobian ill-conditioning we analyze, so that Theorem~2 applies and the condition number $\kappa$ of the local curvature operator becomes large. Similarly, LayerNorm parameters modulate the scale of the representations that feed into attention and the output softmax, thereby controlling how sharply these softmax distributions concentrate and, in turn, how strongly the ill-conditioning we characterize manifests in practice.
>
> **W6 (Proof of Proposition 1 is missing)**: We apologize for the confusion. Proposition 1 is actually
> a special case of our main result, Theorem 2. When the decoupling assumption
> $K_i K_j^\top = 0$ for $i \neq j$ holds, Theorem 2 reduces directly to Proposition 1.
>
> Specifically, under decoupling:
>
> - The cross-example coupling terms in $H_p$ vanish
> - Each example’s contribution decomposes independently
> - The condition number bound of Theorem 2 specializes to give
>   $$
>   \kappa_i \geq \frac{\mu_i}{L_i}
>   \left(
>       \frac{\max_j (a_i^\star)_j}{\min_j (a_i^\star)_j}
>   \right)^2
>   $$
>
> **W7 (Typos and formatting issues in proofs)**: Thank you for pointing out the typos and formatting issues in the proofs, we have addressed them in the revision.

---

> > ### Author Response · Authors · 2025-12-03
> > **Response to Reviewer SdGP - Answers to Questions and Clarifications [Part 1]**
> >
> > **Q1–Q2 (Assumption B validity and prompt tuning evidence):** These are addressed comprehensively in our responses to W3 and W4 above.
> >
> > **Q3 (Hyperparameter tuning for Figure 3):**  This is addressed in our response to W1.
> >
> > **Q4 (Token partitioning for Lemma 3):**  In Lemma 3, the split $(S_i, S_i^c)$ is a modeling device rather than a new algorithmic ingredient: any partition that captures which tokens are task-relevant for example $i$ can be used, and our bounds then hold with constants that depend on this choice (through $s_{\min}, s_{\max}, \Delta_i, \Gamma_i$).
> >
> > In practice, one can define $S_i$ post hoc from a trained model by, for instance, thresholding the learned attention $a_i^\star = \phi(K_i p^\star)$ (taking tokens with $a_{ij}^\star$ above a fixed fraction of the maximum) or by taking the top-$k$ tokens by attention weight.
> >
> > In theoretical setups, the same notion can be grounded in the task structure: for next-token prediction, $S_i$ would consist of tokens in syntactic/semantic relation to the current position, while in a teacher–student setting $S_i$ can be taken as the set of tokens attended by the teacher.
> >
> > The key point is that Lemma 3 expresses our condition-number bounds in terms of whatever partition is chosen: different reasonable choices change the tightness of the constants but not the qualitative message that highly concentrated attention (small $\lvert S_i \rvert$ with a large score gap to $S_i^c$) leads to small $a_U$ and hence poor conditioning.
> >
> > **Q5 (Derivation of Equation 39):**  The equation follows from a direct application of the product rule. Indeed, since
> > $$
> > J(x) = \operatorname{diag}(x) - x x^\top,
> > $$
> > we can write
> > $$
> > J(x) f(x)
> > = \operatorname{diag}(x) f(x) - x \big(x^\top f(x)\big).
> > $$
> >
> > For the first term, using the componentwise product rule on
> > $[\operatorname{diag}(x) f(x)]_i = x_i f_i(x)$ gives
> > $$
> > \frac{d}{dx}\big(\operatorname{diag}(x) f(x)\big)
> > = \operatorname{diag}(x) D f(x) + \operatorname{diag}\big(f(x)\big).
> > $$
> >
> > For the second term, let $s(x) = x^\top f(x)$ so that the term is $x s(x)$.
> > Then $\nabla s(x) = f(x) + D f(x)^\top x$, and another product rule yields
> > $$
> > \frac{d}{dx}\big(x s(x)\big)
> > = s(x) I + x \nabla s(x)^\top
> > = (x^\top f(x)) I + x f(x)^\top + x x^\top D f(x).
> > $$
> > Subtracting these two derivatives and regrouping terms gives exactly the desired identity. We have included this explanation in the proof of the revised version.
> >
> > **Q6 (Discarding three terms):**  We have clarified this in the revision by adding:
> > “Our analysis focuses on stationary points that globally minimize $\ell$, and hence also $\tilde{\ell}$, so that $\nabla \tilde{\ell}(a^\star)=0$ and the last three terms in the previous derivative expression vanish.”

---

> > > ### Author Response · Authors · 2025-12-03
> > > **Response to Reviewer SdGP - Answers to Questions and Clarifications [Part 2]**
> > >
> > > **Q7 (Relation to previous Hessian analyses):**
> > >
> > > *Relation to Hessian analyses [2,3] and novelty.*
> > > Zhang et al. [2] and Ormaniec et al. [3] show, at the full-Transformer level, that the Hessian is highly heterogeneous across parameter blocks and that the self-attention Hessian can be written in terms of *attention moments*. Our work is fully consistent with these observations, but addresses a different question: in a minimal, analytically tractable setting (frozen $W,v$, only the prompt/CLS vector $p$ trainable), we *isolate and exactly quantify* how sparse attention alone can create an exponentially ill-conditioned Hessian block and hence lead to the poor performance of SGD.
> > >
> > > Formally, in our setup the Hessian seen by gradient descent on $p$ is
> > > $$
> > > H_p =\sum_{i=1}^N K_i^\top J(a_i^\star)\,H_i J(a_i^\star)K_i,
> > > \qquad
> > > J(a)=\mathrm{diag}(a)-aa^\top,
> > > $$
> > > which is precisely the Hessian block w.r.t. the prompt parameters. This already fits into the Hessian decompositions of [2,3], but our novelty is that we obtain *explicit, margin–dependent bounds*:
> > > $$
> > > \kappa(H_p) \gtrsim \Big(\tfrac{a_I}{a_U}\Big)^2  \sim\ \exp(2\Delta_{\min}),
> > > $$
> > > so that the condition number explodes as attention becomes sparse (small $a_U$ / large margin $\Delta_{\min}$). In contrast, [2] empirically documents block heterogeneity in large models and analyzes generic quadratic surrogates, but does not identify a concrete geometric control parameter (such as the attention sparsity or margin) that provably drives an exponential blow-up of $\kappa$.
> > >
> > > *Attention simplex vs. attention moments in [3].*
> > > Ormaniec et al. [3] express the self-attention Hessian in terms of centered first–to–third moments of each attention row and observe that near one-hot attention leads to highly anisotropic curvature. Our “attention simplex” analysis makes this connection explicit and quantitative. The Jacobian
> > > $$
> > > J(a) = \mathrm{diag}(a)-aa^\top = \mathrm{Cov}_{j\sim a}(e_j)
> > > $$
> > > is exactly the second central moment of the categorical distribution with probabilities $a$, i.e., one of the attention moments in [3]. By working directly on the simplex, we show that as $a$ approaches a vertex (sparse attention) the smallest non-zero eigenvalue of $J(a)$ shrinks proportionally to the smallest attention mass, which in turn causes the smallest eigenvalue of $H_p$ to vanish and the condition number to blow up. Thus, our theory can be viewed as a *sharp, low-dimensional instantiation* of the attention-moment picture in [3]: we translate their qualitative statement “near one-hot attention ⇒ degenerate Hessian” into a concrete, margin-dependent convergence-rate bound for (S)GD in a setting where the dynamics can be solved exactly.
> > >
> > > **Q8 (Relation to Zhao et al.):** Addressed in W5 above.

---

### Author Response · Authors · 2025-12-03
**Area Chair Summary**

We thank the Area Chair for their time and careful consideration, particularly given the additional challenges this review cycle has presented.

**Overview of Scores and Reviewer Engagement:**

This submission received scores of **6, 2, 2, 2** from four reviewers. We respectfully draw the AC's attention to a notable disparity in engagement depth that we believe merits careful consideration.

**Reviewer SdGP (Score: 6, Confidence: 3)** provided a thorough, technically substantive review: checking mathematical details, raising precise questions about derivations (e.g., Eq. 39), and requesting specific clarifications. SdGP expressed willingness to raise their score if concerns were addressed; we have done so in the revision (experimental details added in App. K, additional experimental results to verify Assumption B included in App. J, and proof typos corrected).

**Reviewers uRJn, 6qvD, and 4xQP (Scores: 2, 2, 2)** submitted shorter reviews with lower confidence. As documented below, several of their central criticisms appear to stem from misreadings of the paper's actual content.

**Addressing Specific Reviewer Concerns:**

| Reviewer | Reviewer Claim | Clarification | Evidence |
| :--- | :--- | :--- | :--- |
| **uRJn** | "Are Fig 2,3 conducted based on the tunable token setting?" (suggesting toy setup) | Figures 2–3 train **full Transformers** with **all parameters** updated. | **Lines 432–475** explicitly describe ViT (MNIST) and Transformer (WikiText-2) with full training. |
| **6qvD** | Simplification is "far from practice" | (1) Setting matches prompt tuning.<br>(2) **Joint training** yields same result. | **New Appendix J:** Ablation jointly training $W$ and $v$ confirms stagnation persists. |
| **6qvD** | "Section 3.1 seems less relevant" | Section 3.1 is **structurally essential**: it establishes why standard overparameterized analysis fails. | **Lines 162–215** explicitly frame Section 3.1 as the baseline motivating Section 3.2. |
| **uRJn** | Requests quantitative "theory vs. experiment" overlays | Qualitative regime matching is standard; exact rate overlays are not. | See Wu et al. (2023); Tarzanagh et al. (2023); Deora et al. (TMLR). |
| **4xQP** | "No theoretical result… why Adam would converge faster" | Scope is explaining **why GD fails** (softmax ill-conditioning), not analyzing Adam. | Introduction & Conclusion explicitly define this scope. |

**Alignment with Field Standards:**

We wish to contextualize the expectations expressed by the lower-scoring reviewers.

*   **On simplified models:** Our single-tunable-token analysis follows a well-established methodology for attention dynamics (e.g., **Oymak et al., ICML 2023**; **Tarzanagh et al., NeurIPS 2023**; **Zhang et al., NeurIPS 2024**), where simplified settings yield closed-form dynamics that are then validated on full architectures. Our assumptions are no more restrictive than those in prior accepted work.
*   **On experimental validation:** Our experiments span synthetic validation of Theorems 1–2, ViT on MNIST (6 layers, 4 heads), Transformer language modeling on WikiText-2, and a linear attention ablation demonstrating that the slowdown disappears without softmax. This breadth—synthetic, vision, language, and ablation—exceeds that of typical theory papers in this area.

**Summary of Contributions:**

1.  **Mechanistic explanation for GD stagnation:** We prove that the softmax Jacobian acts as a preconditioner; when attention becomes sparse, the condition number scales as $\Omega(a_I^2 / a_U^2)$, growing exponentially and causing GD to stall.
2.  **Unified theoretical analysis:** We show that standard overparameterized analysis (Theorem 1) breaks down precisely as the model approaches sparse attention, motivating our underparameterized simplex analysis (Theorem 2).
3.  **Empirical confirmation:** All theoretical predictions are validated: GD stalls under sparse attention, converges under uniform attention, and the slowdown vanishes with linear attention.

As Reviewer SdGP notes, *"The paper studies a question that is important for the Transformer optimization community in a very interesting setting"* and *"Insights from the paper can have practical implications."*

We hope the above clarifications are useful in contextualizing the reviews. We believe the technical contributions and empirical validation stand on firm ground, and we trust the AC will weigh these factors appropriately in the final assessment.

---

### Meta-Review · Area_Chair_BxYo · 2026-01-06

**Summary:**

The goal of paper is to try to answer the question of why attention-based models do not train well with (S)GD. The paper provides new theoretical perspectives and experimental justifications. The reviewers have following concerns.
1. Experimental setup should be more detailed.
2. Experimental results are not strong enough to validate the theory.
3. The theoretical focus on prompt tuning is less practical.
4. Only analyze why SDG does not work, but do not provide analysis why Adam works.
5. Two reviewers complain about the writing is hard to follow sometimes.

**Reviewer Concerns:**

1 is easily addressed. Other concerns are somewhat subjective. I think the theoretical analysis in the paper has the potential to deepen our understanding on LLM training, but its current results seem not convincing enough.

**Reviewer Scores:**

6, 2, 2, 2

---

### Decision · Program_Chairs · 2026-01-26

Reject